# Asynchronous Perception Machine for Efficient Test Time Training

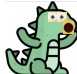

**Rajat Modi**[*]   **Yogesh Singh Rawat**
Centre for Research in Computer Vision
University of Central Florida
Orlando, FL 32765
`rajat.modi@ucf.edu,yogesh@crcv.ucf.edu`
https://rajatmodi62.github.io/apm_project_page

## Abstract

In this work, we propose Asynchronous Perception Machine (APM), a computationally-efficient architecture for test-time-training (TTT). APM can process patches of an image one at a time in any order *asymmetrically,* and *still encode* semantic-awareness in the net. We demonstrate APM's ability to recognize out-of-distribution images *without* dataset-specific pre-training, augmentation or any-pretext task. APM offers competitive performance over existing TTT approaches. To perform TTT, APM just distills test sample's representation *once*. APM possesses a unique property: it can learn using just this single representation and starts predicting semantically-aware features.

APM demostrates potential applications beyond test-time-training: APM can scale up to a dataset of 2D images and yield semantic-clusterings in a single forward pass. APM also provides first empirical evidence towards validating GLOM's insight, i.e. if input percept is a field. Therefore, APM helps us converge towards an implementation which can do *both* interpolation and perception on a *shared-*connectionist hardware. Our code is publicly available at this link.

## 1 Introduction

In these past centuries, computing-machines have become a lot faster [92]. This made it possible to train higher-parameterized neural nets and led to interesting emergent abilities [80]. As was predicted by Turing himself, and as were his suspicions of Lady Lovelace's arguments *against* learning machines[90]: neural-nets can now finally learn without human-feedback [6], paint pictures [76] and even compose a sonnet [92]. Even with such impressive-progress, a key question still remains: how can these nets recognize images whose distribution is far different from the ones which were used during training? For e.g., consider a self-driving car trying to stop when it encounters a pedestrian crossing a road. Such practical scenarios require 'instantaneous-decisions'[2] for ensuring human-safety in autonomous-systems [64].

Test-time-training (TTT) [85] is one of the promising techniques for handling such distribution shifts: a neural net adapts to a test sample 'on the fly'. Since the label of the sample is not known, the net performs some auxiliary pre-text task like data augmentation [15], rotation [15, 86] or prompt tuning [84] on it. After several such iterations, the net recognizes the test sample. The key idea is that the net is allowed to *dynamically* adjust its decision boundary even *after* it has been trained, thereby bringing it much closer to how humans keep learning 'continuously' throughout their lifespans [86].

---

[*]Correspondence to `rajatmodi62@gmail.com`.

[2]A flick of the eye, the fall of her hair, the passing of a second, or capturing the world at 120fps: instantaneous can be a subjective experience. Seconds can sometimes pass slowly for other reasons also.

38th Conference on Neural Information Processing Systems (NeurIPS 2024).

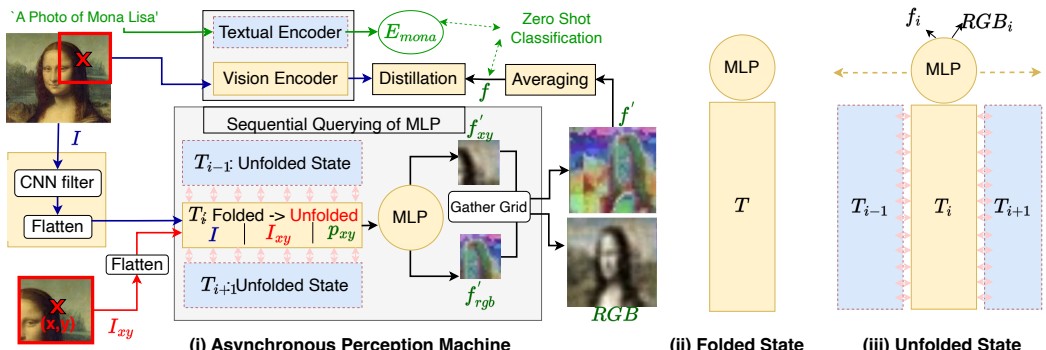

**(i) Asynchronous Perception Machine**  **(ii) Folded State**  **(iii) Unfolded State**

Figure 1: **(i) Asynchronous Perception Machine (APM)**: An image $I$ passes through a column module and routes to a trigger column $T_i$. $T_i$ then unfolds and generates $h \times w$ location-specific queries. These queries are i.i.d and can be parallelized across cores of a gpu [48]. Each query $T_i$ is passed through a *shared* MLP and yields the vector $f'_{xy}$ and $f'_{rgb}$. MLP is queried iteratively until whole grid $f'$ comes into existence. Classification then involves comparing the averaged representation $f$ with class-specific textual representations in the contrastive space . **(ii) Folded State**: The parameters which the net learns are parameters of $T$ and MLP. **(iii) Unfolded State**: $T$ expands to yield $h \times w$ queries 'on-the-fly'. Learning involves oscillating this net between folded and unfolded states. This net can be trained via backpropogation [78, 28].

Despite the success of existing TTT approaches, several limitations need to be addressed [84, 15, 86]: 1) **The Information Bottleneck Problem** [64]: Multiple TTT iterations requires feed-forward through many hidden layers *multiple times*, making it computationally expensive. 2) **Reliance on a surrogate pre-text task:** the optimal data-augmentation pipeline or the best pretext task is not known beforehand, worsening the issue even further in an online setting. 3) Furthermore, TTT leverages architectures like transformers which rely on **parallel perception**: this requires projecting all input patches into a shared representational space, thereby consuming significant memory.

**Inspired by GLOM's philosophy** [29], we hereby propose Asynchronous Perception Machine (APM), a new architecture for efficient test-time-training. 1) It handles the information-bottleneck problem by directly learning a shortcut from input image to final representation from last layer of a model [28]. 2) During TTT, we compute test-sample's representation *only once*. Subsequent iterations involve over-fitting on this representation only and *doesn't* require any data-augmentation/pretext task. 3) APM can operate on a *single* patch at any location *asynchronously* [91] and still encode semantic-awareness in the net, thereby offering a *fresh perspective* towards machine-perception.

We make the following contributions in this work:
- We propose APM, a GLOM-inspired architecture that can perform test-time-training *without* requiring data augmentation/auxilary-tasks. APM is a step towards validating GLOM's philosophical insight: a percept is *really* a field [29].
- APM is computationally efficient, i.e it almost *halves* the number of FLOPs over existing methods [15, 86]. APM matches/surpasses TTT performance by $0.4\% - 8\%$ across 16 datasets.
- APM is architecturally simple: a convolutional layer and a single MLP of 5 layers. APM can still learn using one sample and semantically clusters on a collection of images [28].

## 2  Background

We draw from insights previously philosophically mentioned in GLOM [29]. Consider how machine perception has been done classically: an input $x \in \mathbb{R}^{c \times h \times w}$ is transformed by a non-linear function $f$ to a perceptual feature grid $y \in \mathbb{R}^{c' \times h \times w}$, $c$ denotes image channels, $h, w$ are input spatial dimensions.

**Islands of agreement [29]** Rather than viewing this matrix $y$ as a cuboidal grid, one can now see this as column vectors $v_{c'}$ at each location. There are $h \times w$ such columns in $y$. Therefore, there exists a one-to-one mapping, between each input patch i.e. $(x, y)$ with the vector $v_{c'}$ at that location. A neural net $f$ can then thus learn to fit $f(I, x, y) \to v_{c'}$. These vectors $v_{c'}$ have been termed islands of agreement in GLOM [29]. They were recently demonstrated on realistic datasets by [64], showing some promise that trained models can serve as free supervision-sources for vectors $v_{c'}$.

Next, we imagine feed-forwarding one patch $(x, y)$ into the neural net $f$ and estimating the response $v_{c'}$. It thus becomes possible to process these patches one-by-one rather than keeping them all together in memory [94]. We then further imagine auto-regressively querying $f$ until the whole grid $y$ has been brought into existence.

**Implicit-representations/Neural-Fields:** The neural net $f$ inputs a coordinate based query, $(I, x, y)$ where $(x, y)$ is the 2D location in an image $I$. It then gives the answer $v_{c'}$ [29]. Such implicit representations have been studied extensively in 3D novel-view synthesis [60]. However, in the above problem $I$ is a simple 2D image. In a recent work [29], it was hypothesized that neural-implicit representations can work on 2D images as well without having to retrain it every-time.

**Layer Skipping [28]**: We now combine the previous two insights together. Imagine that $v_{c'}$ has been estimated from a model after travelling through several layers. We can distill these $v_{c'}$ into an implicit representation $f$ and learn a direct mapping between input $x$ and last layer of a model [28]. During inference, we can skip feed-forwarding through *all* the model layers. The only feed-forward cost we then pay is what it costs to travel through $f$. Provided that the number of parameters in $f$ are lesser than parameters in the teacher model, *we can expect computational speedups.*

We now introduce a connectionist-net $f$ motivated by these insights. It can perform efficient test-time-training on a given 2D input.

## 3 Asynchronous Perception Machine (APM)

APM processes an input image $I$ via two novel mechanisms: 1) A column module $T$ which is said to contain an input image $I$, 2) a column folding-unfolding mechanism that operates during each forward-pass. We first provide a technical analogy to better understand APM.

### 3.1 A Technical Analogy[29]

A neural field does 3D novel view synthesis by querying an MLP with a point $(x, y, z)$ and yielding corresponding rgb. In a similar way, APM does *2D perception* by querying an MLP with an image $I$, and a location $(x, y)$ to yield location-specific feature $f_{xy}$. APM features a new mechanism to query the MLP, i.e *a column module $T$*. Next, we define this column representation $T$.

### 3.2 Global Column Module: Defining compressed representation T

We define a column $T$ as a vector of dimensions $1 \times 1 \times d$. Our aim is to map image $I$ in this $T$, so that $T$ can summarize its entire identity. Given an image of dimensions $c \times h \times w$, we run a 2D convolution on it. Number of filters are set as accordingly. The resultant $1 \times h' \times w'$ feature map is then flattened into a single column vector T of dimensions $d = h' \times w'$. We shall refer to this column T as "triggering hyper-column"(seed). *The only learnable parameters in this column are parameters of a convolution filter.*[3] [92, 91].

### 3.3 Abstract view: The Column Unfolding-Folding Mechanism

The trigger column T now starts undergoing cycles of folding-unfolding (Fig 1). During unfolding, the column T copies itself to yield $h \times w$ *location-aware* columns. During folding, these $h \times w$ columns collapse back into a single column $T$. The neural-net then *oscillates* between these folded-unfolded phases during learning iterations.

### 3.4 Computational Principle: Location-aware columns and their collapse

**The unfolding-process** shall now be concerned with generating location-aware column $T_{ij}$ from $T$. We generate $h * w$ 2-D *non-parametric* positional encoding similar to the ones being used in transformers [94] and neural fields [60]. The trigger column $T_{ij}$ is then given by $T_{ij} = (T|p_{ij})$, where $|$ denotes the stacking operator and $(1, 1) \leq (i, j) \leq (H, W)$. T can be said to encode *identity* of an image.

**The folding-process** involves collapsing all columns $T_{ij}$ back into $T$ from which they had begun. This is achievable since the $p_{ij}$ used in $T_{ij}$ was deterministic. An abstract-mathematical intuition on folding is that *all* $p_{ij}$ in $T_{ij}$ get deleted/annihilated at the end of every forward-pass, and only $T$ is left. Positional-encodings contain periodic-sinusoids which offers a strong positional-prior in

---

[3]The word triggering has been chosen because this column shall be used to trigger the queries to the neural field shared across different locations [29].

practice [94]. Gradients collected from all $T_{ij}$ then update the parameters of the convolutional filter in the column $T$. This sharing of T across different locations $(i, j)$ now induces a new fundamental behavior [29] already hypothesized in GLOM (more details in section 5.3).

This representation $T_{ij} = (T|p_{ij})$ exhibits a strong-symmetry breaking behavior [29][4]. For e.g., consider different locations across *same* image $I$. Although the identity T will be the same, the positional encoding $p_{ij}$ shall *differentiate* among different locations. The converse holds true: given *the same* position $p_{ij}$ in two *different* images, the column $T$ shall *differentiate* among identities of different images.

All the experiments in this paper are performed *without-injecting* the local patch $I_{xy}$ in the trigger column $T_{ij}$. This showcases the strong-symmetry breaking behaviour of positional-encodings[94]: they can disentangle location-information from *global I without-requiring* an additional patch-prior.

### 3.5 Firing location-aware columns into a shared MLP

Each column $T_{ij}$ is passed through an MLP to yield location-specific features $f_{ij}$ and RGB values $RGB_{ij}$. Number of neurons in the first layer of the MLP is same as dimensionality of column $T_{ij}$.

**Column Independence:** The MLP is shared across all the columns. One column is also independent of another as illustrated in Fig1. Therefore, the MLP can be queried *sequentially*. Firing a column $T_{ij}$ into the MLP yields a column vector $v_c$. Once $h \times w$ columns are finished firing, we get a feature grid $f$ of dimensions $h \times w \times c$. Note that the number of columns can be as low as 1.

### 3.6 Training and Losses

We detail how the $t^{th}$ iteration of test-time-training could be performed. First, the obtained feature grid $f \epsilon \mathbb{R}^{h \times w \times c}$ from APM is averaged to yield $f_{avg} \epsilon \mathbb{R}^c$. For the first TTT iteration, i.e. $t = 1$, the image $I$ is feed-forwarded through a multi-modal teacher like CLIP to get a CLS token $f_{cls}$ and corresponding text representation $t_{cls}$. APM then learns to estimate this *same* target feature $f_{cls}$. We enforce this by a simple $L_2$ constraint as:

$$L_i = L_2(f_{avg,t}, f_{cls}) \tag{1}$$

where $f_{avg,t}$ is the averaged output feature from APM at a particular TTT iteration t. Note that **the target $f_{cls}$ is only estimated in $t = 1$ and remains the same for $t \geq 2$,** i.e. subsequent feed-forward through teacher is *not* needed.

**Memory-efficient estimation of $f_{avg,t}$:** During a TTT iteration $t$, $f_{avg}$ is computed as a simple average of $f \epsilon \mathbb{R}^{h \times w \times c}$. This would require $h \times w$ columns to exist in the memory *simultaneously*. APM's design assumes column independence which allows estimating $f_{avg}$ as a statistical running average, i.e.

$$f_{avg} = \frac{n \times f_{avg} + f_{i,j}}{n + 1} \tag{2}$$

assuming, $n$ columns have already been fired into the APM and one additional column corresponding to position $(i, j)$ of image $I$ is in the process of firing. This procedure is repeated until all positions $(i, j)$ are exhausted. We represent the sequential column-firing by a 'Gather-Grid' operator in Fig1.

**Predicting image class-label:** After certain TTT iterations, (say t = 20), the output feature $f_{avg,t}$ and the textual features $t_{cls}$ are obtained. Image-classification then follows the standard practice of comparing the distance of $f_{avg,t}$ with each plausible class feature $t_{cls}$ in the contrastive space and choosing the closest one as the prediction [75].

## 4 Experimenting with APM

We quantitatively evaluate APM on *zero-shot* test-time-training on popular benchmarks containing distribution-shifts [15, 86, 84]. Next, we quantitatively explore its computational efficiency.

**Datasets:** Cifar-10C [86] contains 5 level of corruptions on the test-set with 15 types of noises. *Larger* datasets with significant distribution shifts consists of ImageNet val/other curated splits. For e.g., **ImageNet-V2** contains natural images consisting of $10k$ images across 1000 classes. **ImageNet-A** contains 7500 adversarial-images consisting of 200 categories. **ImageNet-R** consists of 30000

---

[4]It means generating a unique representation $T_{ij}$ for any image and a location selected in it. Another way to break symmetry is to add noise [40]. Noise also helps escape degenerate local-minima. These ideas can be traced back to boltzmann-machines/hopfield-nets[44].

artistic-images across 200 ImageNet categories. **ImageNet-Sketch** consists of black/white sketches of 50000 images for 1000 classes. **ImageNet-C** consisting of 15 types of corruptions with 5 levels of severity. There are additional 9 cross-dataset generalization datasets [84].

**Baselines:** We compare against standard TTT-online [86], TTT-MAE [15], TPT [84], CLIP VIT-B/16, Coop, CocoOP. We also benchmark CLIP VIT-L/14 and the strongest OpenCLIP VIT-H/14-quickgelu variant pre-trained on *dfn5b*.

Table 1: **APM's Robustness to Natural Distribution Shifts**. CoOp and CoCoOp are tuned on ImageNet using 16-shot training data per category. Baseline CLIP, prompt ensemble, TPT and our APM do not require training data. A ✓ in P means that method leveraged **pre-trained weights** on clean variant of train set aka, Image-net and downstream-ttt on corrupted version.

| Method | P | ImageNet Top1 acc. ↑ | ImageNet-A Top1 acc. ↑ | ImageNet-V2 Top1 acc. ↑ | ImageNet-R Top1 acc. ↑ | ImageNet-Sketch Top1 acc. ↑ | Average | OOD Average |
|---|---|---|---|---|---|---|---|---|
| CLIP-ViT-B/16 | ✗ | 66.7 | 47.8 | 60.8 | 73.9 | 46.0 | 59.1 | 57.2 |
| Ensemble | ✗ | 68.3 | 49.8 | 61.8 | **77.6** | 48.2 | 61.2 | 59.4 |
| TPT | ✗ | **68.9** | **54.7** | 63.4 | 77.0 | 47.9 | 62.4 | 60.8 |
| APM (Ours) | ✗ | 68.1 | 52.1 | **67.2** | 76.5 | **49.3** | **62.6** | **61.2** |
| CoOp | ✓ | 71.5 | 49.7 | 64.2 | 75.2 | 47.9 | 61.7 | 59.2 |
| CoCoOp | ✓ | 71.0 | 50.6 | 64.0 | 76.1 | 48.7 | 62.1 | 59.9 |
| TPT + CoOp | ✓ | 73.6 | 57.9 | 66.8 | 77.2 | 49.2 | 64.9 | 62.8 |
| TPT + CoCoOp | ✓ | 71.0 | 58.4 | 64.8 | 78.6 | 48.4 | 64.3 | 62.6 |
| CLIP VIT-L/14 | ✗ | 76.2 | 69.6 | 72.1 | 85.9 | 58.8 | 72.5 | 71.6 |
| APM (Ours) | ✗ | **77.3** | **71.8** | **72.8** | **87.1** | **62.2** | **74.2** | **73.4** |
| OpenCLIP-VIT-H/14 | ✗ | 81.6 | 79.1 | 80.7 | 92.9 | 72.8 | 81.4 | 81.3 |
| APM (Ours) | ✗ | **84.6** | **84.2** | **83.9** | **94.9** | **77.1** | **84.9** | **85.0** |

Table 2: **APM's performance on ImageNet-C, level 5**. The first three rows are fixed models without test-time training. The third row, ViT probing, is the baseline used in [15]. A ✓ in P means that method leveraged **pre-trained weights** on clean variant of train set aka, Image-net and downstream-ttt on corrupted version. CLIP VIT-L/14 is generally more robust. APM does better on $11/15$ noises with an average accuracy score of 50.3.

| | P | brigh | cont | defoc | elast | fog | frost | gauss | glass | impul | jpeg | motn | pixel | shot | snow | zoom | Average |
|---|---|---|---|---|---|---|---|---|---|---|---|---|---|---|---|---|---|
| Joint Train | ✓ | 62.3 | 4.5 | 26.7 | 39.9 | 25.7 | 30.0 | 5.8 | 16.3 | 5.8 | 45.3 | 30.9 | 45.9 | 7.1 | 25.1 | 31.8 | 24.8 |
| Fine-Tune | ✓ | 67.5 | 7.8 | 33.9 | 32.4 | 36.4 | 38.2 | 22.0 | 15.7 | 23.9 | 51.2 | 37.4 | 51.9 | 23.7 | 37.6 | 37.1 | 33.7 |
| ViT Probe | ✓ | 68.3 | 6.4 | 24.2 | 31.6 | 38.6 | 38.4 | 17.4 | 18.4 | 18.2 | 51.2 | 32.2 | 49.7 | 18.2 | 35.9 | 32.2 | 29.2 |
| TTT-MAE | ✓ | 69.1 | 9.8 | 34.4 | 50.7 | 44.7 | 50.7 | 30.5 | 36.9 | 32.4 | 63.0 | 41.9 | 63.0 | 33.0 | 42.8 | 45.9 | 44.4 |
| OpenCLIP VIT-L/14 | ✗ | 71.9 | 47.0 | 50.3 | 32.7 | 58.3 | 46.9 | 26.0 | 26.5 | 28.1 | 62.7 | 37.7 | 58.3 | 28.2 | 50.4 | 37.9 | 42.1 |
| APM (Ours) | ✗ | **77.4** | **51.9** | **56.6** | **37.9** | **64.8** | **53.2** | **28.7** | **31.4** | **33.0** | **68.4** | **44.1** | **64.5** | **33.1** | **56.9** | **43.9** | **50.3** |

**Results and Analysis:** We study APM's performance on test-time-training on several datasets. APM processes each test sample individually: i.e. the weights are drawn from a normal distribution after processing every sample to prevent information leakage. For zero-shot classification of a test-sample, APM leverages the 80 textual-prompt templates similar to the ones used in CLIP. In Tab 1, APM scales up to *zero-shot* classification task to datasets with 1000 classes. Using CLIP-ViT B/16 as a teacher, we surpass TPT [84] with an avg score of 62.6 and avg ood-score of 61.2. Next, we benchmark OpenCLIP-VITH/14 against all these splits. Using the same model as our teacher, we get an absolute improvement of 3% ↑ [84.6%] on ImageNet val set, 5.1% ↑ [84.2%] on ImageNet-A, 3.2% ↑ [83.9%] on ImageNet-V2, 2% ↑ [94.9%] on ImageNet-R, 4.3% ↑ [77.1%] on ImageNet-Sketch respectively. A similar trend is observed with a VIT-L/14 backbone. This *might* lead to the conclusion that a stronger teacher seems to benefit APM. In Table 2, we show results on Imagenet-C. Using a CLIP VIT-L/14 teacher, APM gets highest accuracy on $11/15$ noises with the highest average score of 50.3. Note that TTT-MAE also uses a VIT-L encoder, and is $pretrained$. In contrast, $APM\ doesn't$ need any dataset specific-pretraining. Finally, in Tab 3, APM improves upon $4/9$ datasets, comes close on remaining 5 and gets the highest average accuracy score of 65.5.

**APM is computationally efficient:** All experiments are run on a *same* desktop-workstation containing 1x rtx a6000/96GB ram/Ubuntu 22.04/2TB ssd. Flops were counted with meta's fvcore library [89]. In Tab4, we perform 20 TTT iterations. TTT on CLIP-VIT B/16 baseline used in [84] consumes 462Gflops. Next, we do TTT leveraging APM. At $t = 1$, feed-forwards involves clip-teacher and consumes 20.5 flops. Remaining 19 TTT iterations only involve overfitting on distilled image token at $t = 1$, and consumes 10 flops/TTT-iteration. The entire profile-dump yields 241.7 flops, which is an almost 50% reduction over 462 flops. The total memory occupied by APM i.e. $2.7GB$ is more than CLIP-VIT/B $2.3GB$ since teacher is kept in memory during TTT. However, APM only occupies $600MB$ and reduces actual consumed-flops by 50%.

Table 3: **Cross-dataset generalization** from ImageNet to fine-grained classification datasets. CoOp and CoCoOp are tuned on ImageNet using 16-shot training data per category. Baseline CLIP, prompt ensemble, TPT and APM do not require training data or annotations. We report top-1 accuracy.

| Method | P | Flower102 | DTD | Pets | UCF101 | Caltech101 | Food101 | SUN397 | Aircraft | EuroSAT | Average |
|---|---|---|---|---|---|---|---|---|---|---|---|
| CoOp | ✓ | 68.7 | 41.9 | 89.1 | 66.5 | 93.7 | 85.3 | 64.2 | 18.5 | 46.4 | 63.9 |
| CoCoOp | ✓ | 70.9 | 45.5 | 90.5 | 68.4 | 93.8 | 84.0 | 66.9 | 22.3 | 39.2 | 64.6 |
| CLIP-ViT-B/16 | ✗ | 67.4 | 44.3 | **88.3** | 65.1 | 93.4 | 83.7 | 62.6 | 23.7 | 42.0 | 63.6 |
| Ensemble | ✗ | 67.0 | 45.0 | 86.9 | 65.2 | 93.6 | 82.9 | 65.6 | 23.2 | 50.4 | 64.6 |
| TPT | ✗ | **69.0** | 47.8 | 87.8 | 68.0 | **94.2** | **84.7** | 65.5 | 24.8 | 42.4 | 65.1 |
| APM (Ours) | ✗ | 62.0 | **48.9** | 81.6 | **72.6** | 89.6 | 84.2 | **65.7** | **29.7** | **55.7** | **65.5** |

Table 4: **APM's computational analysis**: TTT for 20 iterations on APM. Baseline is CLIP VIT-B/16 which is used as a teacher in [84]. $M_{meas}(GB)$, $GFlops_{meas}(GB)$ are tmeasured stats. $M_i(GB)$, $GFlops_i(GB)$ are idealistic estimates. (t): $t^{th}$ ttt iteration, (s): student, (u): teacher, (s+u): portion of memory/flops consumed by student/teacher respectively. Note that APM is a $25M$ net.

| | $t$ | Params(M)$\downarrow$ | $M_{meas}(GB)\downarrow$ | $M_i(GB)\downarrow$ | $GFlops_{meas}\downarrow$ | $GFlops_i\downarrow$ |
|---|---|---|---|---|---|---|
| Swin[52] | 1-20 | 87 | 1.5 | 1.4 | 353 | 308 |
| TPT[84] | 1-20 | 151.3 | 3.1 | 2.7 | 529 | 476 |
| CLIP VIT-B/16 | 1-20 | 149.2 | 2.3 | 1.8 | 462 | 410 |
| CLIP VIT-B/16(u)[84] | 1 | 149.2 | 1.8 | 1.8 | 20.5 | 20.5 |
| APM(s) | 1 | 174.2(s+u) | 2.7 (s+u) | 1.8(u) + 0.6(s) | 20.5(u) | 20.5(u) |
| APM(s) | 2 | 174.2(s+u) | 2.7 (s+u) | 1.8(u) + 0.6(s) | 10(s) | 10(s) |
| Peak Use | 1-20 | 174.2(s+u) | 2.7 (s+u) | 1.8(u) + 0.6(s) | **241.7 (s+u)** | **210.5 (s+u)** |

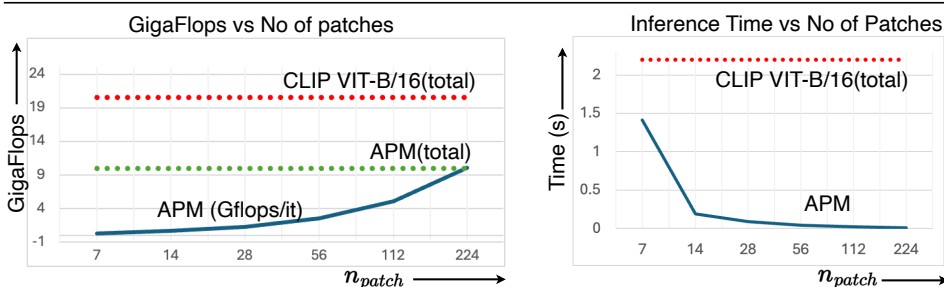

Figure 2: **APM's analysis with variable number of patches:** (left) Gflops of CLIP VIT-B/16 and APM as a function of number of processed patches. (right) Feed-forward time vs number of patches.

**APM does patch-based processing:** In Fig 2, we estimate GFlops/time to process a $224 \times 224$ image using CLIP VIT-B/16 & APM. CLIP VIT-B/16 consumes 20.5 Gflops. On the other hand, APM takes only 10 Gflops in total, in part due to lesser parameters. APM's effectiveness comes from processing as low as 7 patches at the same time: it occupies lesser memory but takes more time (i.e. 1.5 seconds). $1.5sec$ is still lower than VIT-B/16's $2.2sec$. The extreme lies when all patches are processed: Inference time in APM goes down to $0.002$ seconds compared to VIT-B/16's 2.2 seconds, thereby indicating its effectiveness. Note that VIT-B/16 can't do this patch-based processing.

APM's computational effectiveness stems from this *unique* ability to overfit on a *single* test sample's embedding which was distilled only at $t = 1$. This merits a deeper investigation.

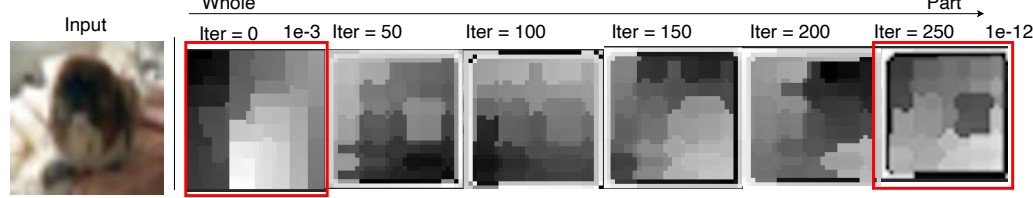

Figure 3: **Overfitting on a *single* distilled token representation leads to islands of agreement[29]:** APM is overfit on a test-sample's representation distilled from a teacher. We plot t-sne clustering of output features over 250ttt iterations. $L_2$ loss between predicted features and distilled sample falls from 1e-3 to 1e-12. Moving left to right shows that wholes break into smaller parts.

**APM can learn on a single sample:** In Fig 3, we feed-forward a single image through our APM and perform 250 TTT iterations[5]. We only *overfit* on one distilled test-sample embedding and plot the 2D

---

[5]A larger number of TTT iterations are needed to perceive semantically-meaningful islands. Actual TTT process is far quicker, it takes around 20 iterations.

t-sne clustering of predicted APM. It can be seen that the elements of the scene have gradually started to cluster. This suggests that APM solves an *inverse* problem: given a single test-sample embedding, what features in the image led to its formation? Over several TTT iterations, APM's features become representative enough to explain different parts of a scene[29]. Therefore, the same-net *could be made to move* up-down the part-whole hierarchy[28], although it requires TTT-iterations *for now*.

Next, we explore if data augmentation *improves* APM's performance. We perform TTT iterations on CIFAR-10's test set and find that data augmentation drops APM's performance from 98.6 to 76.7. This quantitatively demonstrates that APM works best when it does *one-sample* overfitting. It now aligns with the earlier qualitative experiment: over TTT-iterations, the network is learning to cluster the elements in the scene3. Data augmentation *distorts* the sample and makes it difficult for APM's predicted features to *agree* on a stable, *relaxed*-representation that explains the scene[42, 26]

Till now, APM's operation has involved random-initialization of weights for *every* test-sample and performing test-time-training. There is another mode that it can be made to operate in.

## 5 APM Training (Qualitative Analysis)

APM can also scale up and do learning on a *batch* of samples (for e.g., COCO images[50]) distilled from a teacher[31]. This requires introducing several new mechanisms. Note that this section is meant to *qualitatively* demonstrate how scaling up APM can improve the net's interpretability, and help seed future research. Quantitative experiments beyond test-time-training remain *out of the scope* of this paper. APM's training follows a standard setup in self-supervised-learning[6]. We have provided the full algorithm for SSL-training/test-time-training in the AppendixC. During inference, APM takes any 2D image $x_k$ and predicts its RGB reconstruction $RGBk$/higher dimensional features $f_k$. The net then begins to demonstrate several interesting properties, which we will discuss next.

### 5.1 APM can do RGB reconstruction for any 2D input.

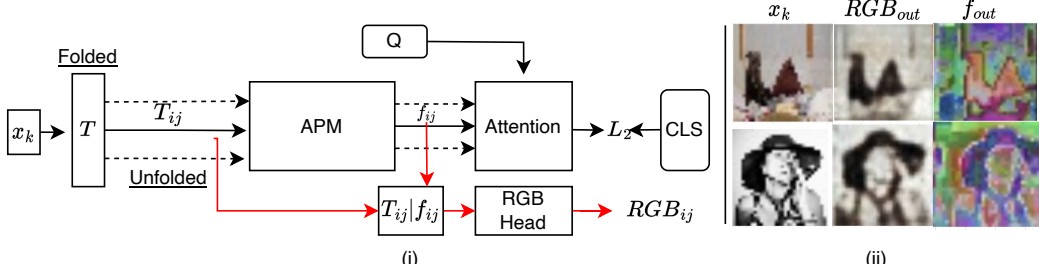

Figure 4: **RGB Decoding in APM**: Input trigger column $T_{ij}$ is concatenated with predicted feature $f_{ij}$ and fed to downstream RGB head. This decodes RGB logit at location (i,j) for *any* 2D input $x_k$. (ii) Input $x_k$ sampled from Coco-val set. $RGB_{out}$: reconstructed RGB, $f_{out}$: Predicted feature grid.

Given a sample $x_k$, APM can reconstruct its RGB. In Fig4, we achieve this by estimating $f_{rgb,ij} = (T_{ij}|f_{ij})$. This skip-connection from trigger column $T_{ij}$ to output feature has a *subtle* reason: consider a *white* dog and *brown* dog. The predicted object-level feature for both will be almost identical[29, 2]. However, $T_{ij}$ is different for both since it contains lower patch-level features[29]. Therefore, this helps us break symmetry. Without this skip-connection[22], the net fails to predict RGB. The network is trained to reconstruct RGB for a batch of images, $L_{rgb} = \sum_{i=1}^{N} \sum_{j=1}^{h*w} L_2(p'_j, p_j)$, where $p_j/p'_j$ are ground truth/predicted RGB-logits respectively.

### 5.2 APM is asynchronous yet encodes semantic-awareness in the net.

Given a sample $x_k$, APM can directly *learn* to mimic the entire *last layer* feature-grid which a teacher model would have generated. We enforce this by a $L_{grid} = \sum_{i=1}^{N} \sum_{j=1}^{h*w} L_2(f'_j, f_{grid})$. In Fig 5(ii) we estimate the error map between the features predicted by our APM and Dinov2[67]. The error map is mostly black which shows that it closely approximates Dinov2's grid[6]. Note that APM does patch-based *asynchronous* processing whereas DINOv2 relies on parallel perception. Finally, Fig 5(iii) shows a simple feed-forward of a CIFAR-10 sample through APM. We can see semantically-aware features. Note that this is a *single* feed-forward through APM[29]. Predicting output feature grid allows the net to encode dark knowledge, i.e. the knowledge of both correct

---

[6]This particular net got a $L_2$ grid loss of 0.15.

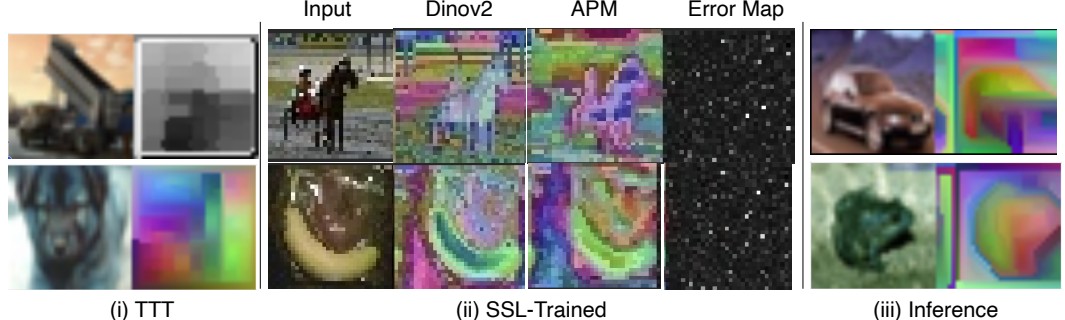

Input     Dinov2     APM     Error Map

(i) TTT             (ii) SSL-Trained             (iii) Inference

Figure 5: **APM feature Analysis:** (i) TTT iterations on an input image leads to semantically aware clustering. top: 2D t-sNE. bottom: 3D t-sNE. [64, 29]. (ii) APM is trained via self-supervision using DINOv2-Teacher. (from left) Input, Dinov2 grid, APM grid. APM's grid **closely approximates** Dinov2 grid evident from black regions in error map. Note that APM does asynchronous patch-based processing whereas Dinov2 does parallel perception. (iii) Cifar-10 samples feed-forwarded through SSL-trained APM yields features of significant semantic quality.[29]

and *incorrect* classes[30]. This is better than just mimicking one hot vector of a target class since it manages to encode lower probabilities of wrong classes also.

### 5.3 APM is a step towards validating GLOM's insight: input percept is a field[36].

Source     1     2     3     4     5     Target

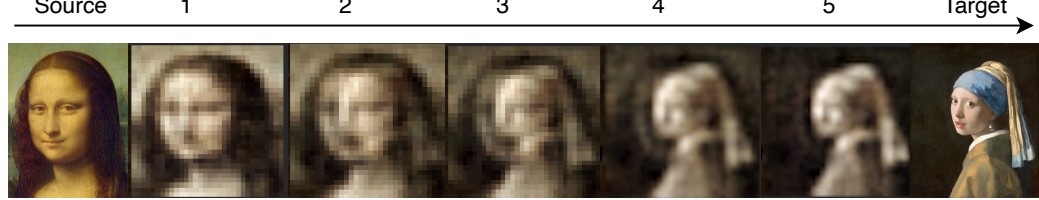

Figure 6: **APM is a step towards validating GLOM's insight [29]**: input percept is a field. An interpolation between *any* two images in the wild. This field arises in APM's MLP consisting of *5 layers*. Trigger column $T$ acts as a key which retrieves an image from the APM's memory. T resides in a continuous embedding space, not discrete addressing space.

In Fig 6, we show that APM can interpolate between any two images in the wild. We choose two images $I_1$ and $I_2$. These images are then funneled through the trigger column $T$ and yield two vectors $v_1$ and, $v_2$ respectively. Next, we generate n intermediate latents separated by an equal linear distance by $v_j = v_1 + \frac{v_2 - v_1}{n}$. Each latent then brings into existence its own set of location-aware columns and decodes an image from the MLP. Such an interpolation has been previously observed in other models [20]. APM now functions as a *new form of addressing mechanism*: the trigger column T acts a key. *Copying* T across locations yields image-specific queries[29]. Values are synapses triggered in the MLP. RGB decoding happens in the output head. Hence, such *continuous* keys and queries exist *outside* the net[29].

Classically, auto-regression has unrolled a shared-decoder over time[98]. In contrast, APM holds the whole sequence $I$ in $T$, and directly hops onto a space/time-step[12] by querying the MLP with a location-column $T_{ij}$. Note that $T_{ij}$ is generated by unfolding $T$. Recurrence/feedback-loops are compensated for by a form of feature-expression[29]. This is a step towards validating GLOM's insight, i.e. input-percept is a field[29] and one can now interpolate in it (gestalt psychology). Furthermore, the trigger column $T$ resides in a continous embedding space, and not discrete hardware locations(classical AI)[29]. Therefore, APM tries to integrate insights from both fields.

## 6 Ablations on APM

The experiments on TTT had relied on a curious ability of APM: it could simply overfit on a test-sample's distilled representation at $t = 1$. This merits further investigation:

**Effect of one-sample test-time training:** In Table 5a, we investigate whether existing networks are capable of one-sample test-time-training. We employ a randomly-initialized network to overfit on a distilled test sample's token obtained in the first TTT iteration [28]. Standard MLP achieves low accuracies of $9.0$ and $3.8$ on CIFAR-10 and CIFAR-100, respectively. Notably, an 11.4M parameter ResNet18 outperforms the larger ResNet34 with 21.5M parameters. A reason might be that too many parameters in ResNet34 gives it too many degrees of freedom[32]: it finds it hard to overfit on one

Table 5: **Ablations on APM**. All nets except CLIP VIT-L/14 use random weights b) $T_c$: trigger column contains convolutions. $T_{vit}$: Trigger column contains a routed VIT representation. C-10: CIFAR-10, C-100: CIFAR-100. Accuracy is reported.

(a) Ablation to evaluate abilities of existing nets to learn from a single sample[22, 77].

|  | Params | C-10 | C-100 |
|---|---|---|---|
| Zeroshot |  |  |  |
| CLIP VIT-L/14 | 428M | 95.37 | 73.28 |
| MLP | 21M | 9.0 | 3.8 |
| ResNet18 | 11.4M | 85.69 | 21.77 |
| ResNet34 | 21.5M | 78.24 | 12.89 |
| APM | 25M | **97.04** | **77.98** |

(b) Ablations of our APM on C-10.$I_{xy}$ means that local patch was injected into the column $T_c$.

|  | $L_{grid}$ | $L_{cls}$ | $L_{rgb}$ | $I_{xy}$ | Acc |
|---|---|---|---|---|---|
| $T_c$ | ✗ | ✓ | ✗ | ✗ | 94.2 |
| $T_c$ | ✗ | ✓ | ✓ | ✗ | 91.0 |
| $T_c$ | ✓ | ✓ | ✗ | ✗ | 96.1 |
| $T_c$ | ✓ | ✓ | ✓ | ✗ | 96.5 |
| $T_c$ | ✓ | ✓ | ✓ | ✓ | **96.8** |
| $T_{vit}$ | ✓ | ✓ | ✓ | ✓ | **98.6** |

sample. Our APM demonstrates strong performance on both CIFAR-10 and CIFAR-100, surpassing the CLIP VIT-L/14 baseline.

**Effect of various losses on APM:** We analyze each row in Table 5b. Initially, only the CLS token from the teacher was distilled into our network, resulting in a yield of $94.2\%$. When we added RGB reconstruction loss for input, accuracy dropped to 91.0, attributed to the difficulty of breaking RGB symmetry [23]. Subsequently, mimicking the entire feature grid and CLS token from the teacher increased accuracy to 96.1. Adding both $L_{rgb}$ and $L_{grid}$ further improved performance to 96.5. $L_{grid}$ here refers to the last-layer of the teacher. Notably, while simple RGB reduction decreased performance (91.0), a combination of $L_{grid}$ and $L_{rgb}$ enhanced our network, as lower RGB and higher object features complement each other [29]. Injecting local patch information $I_{xy}$ into the trigger column improved performance to 96.8. Finally, routing output tokens from a single VIT layer into the trigger column $T_{vit}$ strengthened the column and improved performance.

**Effect of increasing number of convolution in T:** Increasing number of convolutional kernels from 1 to 3 improves from 96.08 to 97.67.

These ablations reveal: 1) APM can do one-sample overfitting for a test sample, 2) It helps to have both local patch and a strong image representation in the trigger column T, and 3) Increasing the levels of part-whole supervision strengthens the net.

## 7 Related Work

**Prompting Approaches:** Prompting is a mechanism to adapt a foundational-model to a downstream task in a zero-shot manner[58]. However, prompting typically requires well-designed hand-crafted prompts. Prompt-tuning methods consist of learnable prompts which enable a parameter-efficient approach to fine-tune a foundational-model. CoOp[106] applied prompt-tuning to CLIP. However, CoOp[106] is sensitive to OOD-data, which CoCoOp[105] in turn compensates for by conditioning the prompts on model inputs. Similarly, TPT[84] optimizes a prompt to encourage consistent predictions across multiple augmented views of the same test sample, and uses confidence-selection to filter out noisy-predictions. Note that TPT[84] performs prompt-tuning over ViT-B/16 but requires feed-forward through *all the model-layers* for *every* iteration.

**Test-Time-Optimization:** Introduces the notion where a model adjusts its decision boundary dynamically during testing, for eg, improving robustness to distribution shifts. Test-time training (TTT) generally adds a self-supervised multi-task branch, and performs a SSL-task like rotation, or masked-reconstruction to adapt the network to the test-sample[15]. These approaches typically initialize the net with pre-trained weights, for eg, Imagenet before undergoing ttt-iterations on a downstream corrupted-dataset. An alternate line of work, for eg, TENT[96] proposes to minimize entropy of batch-wise test-samples. However, TENT[96] requires more than one test-sample to converge towards an optimal solution, whereas APM can also operate on one test sample. Another line of work adjusts internal batch-norm-stats of a network[81]. However, this makes the network-architecture inflexible and requires more than one test sample for optimization. Several other papers following the original pioneering-TTT paper[86] have worked on different problem formulations, for eg, assuming access to an entire dataset (e.g. [51, 72, 97, 103, 18, 104, 19]) or a batch (e.g. TENT [96]) of test inputs.

In contrast, APM evaluates on a each test-sample independently. Inductively, APM does not require any dataset specific pre-training, a pretext task or prompt tuning. Some approaches also use higher-parameterized transformer/diffusion models[72] making the optimization compute-intensive. However in APM, for TTT iteration $t > 1$, feed-forward is done through only 25M parameter APM and not the 149.2M Clip ViT-B/16, resulting in computational efficient test-time-training(Fig 2).

APM also inherits zero-shot behaviour from CLIP, which allows it to bypass training a separate dataset-specific linear-probe for downstream TTT. Finally, some works in areas like source-free domain-adaptation do perform TTT on smaller datasets like Cifar-10 etc[96]. APM additionally shows results on Imagenet splits (Tab1,2) and various cross-generalization datasets(Tab3).

**Part-whole hierarchies:** The idea of encoding part-whole dynamic-parse-trees as distributed representations in neural nets can be traced back to [33], with recent attempts leading to capsule networks[42, 79]. However, the EM[42]/attention-based[79] routing forces *each* capsule to represent only one part[64, 62, 29]. This fundamental flaw prevents capsules from scaling-up and generalizing to multiple OOD objects[64]. Recently, GLOM[29] proposed a *theoretical* system of representing each input pixel as containing a column-vector. These vectors then undergo a routing procedure such that pixels corresponding to same part come to 'agree' with each other.[16] demonstrated GLOM on Cifar splits. APM does not need any routing because it processes location-aware columns independently. APM now reveals an *additional* perceptual-interpolation property not shown earlier(Fig6).

**Knowledge Distillation:** There has been a long history of training data-specific mixture/product of experts and having their ensemble vote towards a prediction, with the key idea to make the experts as different from as each other as possible, and only one expert deciding on the predictions of a particular input sample[46]. This is better than having all experts give equal opinion on a sample[34]. Other methods attempted to 'gather' their collective knowledge to a single model for edge-device deployment[31]. Recently, knowledge has been transferred from a larger teacher model to smaller ones[5], for eg, a foundational model[88, 74]. The student often retains/becomes-better than its teacher[1]. In practice, the weights of teachers are fluctuated slowly (EMA) as compared to the student[87]. This mechanism then helps realize Kahnman's theory of slow-fast thinking[47].

Typically, distillation requires boltzmann-matching[4] predicted distribution between students and teachers, with the distribution's sharpness being governed by a temperature parameter[40]. In contrast, APM directly mimics the entire last layer feature grid of a teacher via $L_2$ norm[93]. Furthermore, APM possesses a novel-inductive bias that *can recover* semantic-features from a single CLS-token distilled from a teacher[Fig 5]. This validates the intuition that CLS tokens encode useful geometric-information of a scene *after* cross-attention of CLS token with patch tokens of an input image[11].

**Routing Mechanisms**: Routing mechanisms involve routing correct object-specific information to correct neurons. [54] proposed slots which perform binding by iterative rounds of self-attention. Löwe et Al[56] proposed 2D complex autoencoder and rotating features[57] where presence of an object is encoded in phase of a neuron and follows the minimum description length principle[39]. In APM, binding is done via the location itself[29]. GroupVIT [101] routes information to multiple group tokens and shows that semantic segmentation emerges with just image-text contrastive supervision.

## 8 Conclusion

We propose APM, inspired from the insights presented in GLOM[29, 38]. APM *promises* to be an efficient architecture for test-time-training and asynchronous patch-processing[91]. APM shows robustness to extreme distribution shifts[24]. APM demonstrates that MLP's can be made to semantically cluster a given image. We hope that APM will help *inspire* further research on *simpler* weight-sharing, lower-memory, higher-bandwidth efficient-nets [28].

**Limitations:** In this work, we have mainly-focused on image-classification. Furthermore, APM requires multiple TTT iterations for now, although they might be reduced by exploring pre-training on a source-dataset[28]. APM can work on just 1 sample with randomly-initialized weights. However, it still requires a single CLS token which has been distilled from a teacher pre-trained on a large-scale dataset. Our preliminary-experiments have revealed that APM can still do RGB-reconstruction *without* a teacher[23], showing potential that APM can be self-sufficient and independent.

## 9 Acknowledgements

This research is supported by the Intelligence Advanced Research Projects Activity (IARPA) via Department of Interior/ Interior Business Center (DOI/IBC) contract number 140D0423C0074. The U.S. Government is authorized to reproduce and distribute reprints for Governmental purposes notwithstanding any copyright annotation thereon. Disclaimer: The views and conclusions contained herein are those of the authors and should not be interpreted as necessarily representing the official

policies or endorsements, either expressed or implied, of IARPA, DOI/IBC, or the U.S. Government. We also thank many other people and MLCollective whose support made this work possible.

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

# Appendix: Asynchronous Perception Machine for Efficient Test-Time Training

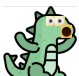

## A  Broader Impact

There are two main ideas that led us to Asynchronous Perception Machine[40, 29]. The first idea is that instead of thinking of features as a cuboidal feature grid[27], one can think of it as a column vector at each location[29, 64]. This helped us learn one to one mapping between the input rgb patch and the vector at a particular location. It led to the net being able to process one patch at a time.

The second idea is this notion of collapsing information into a single starting point. Our previous understanding was that this 'collapse' leads to degeneracy[40]. In this paper, the information can be recovered from the starting point by copying it many times and breaking symmetry with positional encoding. By asking the right questions at the right place at the right time, a net can thus learn to express correct features[91, 29]. Although the information can 'degenerate' to a single starting point, the net can still learn thanks to the strong positional-prior injected by such periodic-encodings[94].

This notion of combining information coming from many locations appears to have connections to Kolmogorov-Arnold superposition theorem[29, 53]. The single convolutional filter in APM might be considered an encoder and five layered MLP as a decoder. Convolution filter can also be considered to be a tape[92] on which symbols are written, processed by a learning machine(MLP)[92] which speaks the correct answers. We have observed that MLP's have this ability to cluster elements in any image. This seems exciting for dense visual tasks with potential for new insights. Finally, we are led to believe that the networks could be made even smaller with higher bandwidth [29, 28]. Of course, it shall mean defining a metric called *bandwidth*, where the performance of the learning machine shall be measured by a three-tuple of <*parameters, accuracy, bandwidth*>[28, 2].

Knowledge-transfer can then be a consequence of sharing folded-embeddings which could grow to form dynamically connected-networks rather than mimicking unfolded-outputs among multiple neural-nets. Knowledge transfer between nets of same structure is as simple as copying weights from one to another thereby making them *immortal*. There already exist approaches which share trees or share knowledge between different neural nets, for e.g. dropout. A higher bandwidth way might involve exchanging folded network-vectors in higher dimensional space, which then reduces to the setting of distributed federated-learning that could run in low-cost hardware[7].

A Turing machine gives us a sense of *closure*[68]: the input tape is shared for both input and output to the machine. However, existing neural nets are mostly bottom-up, with feature expression limited to last layer of the net. In contrast, neurons of a boltzmann machine were clamped, to allow data-vectors to be presented to the net via the environment *as well as* express the generated perceptual codes[14] sampled from the net itself on a same set of neurons[35, 37]. Modelling this closure presents problems for training a neural net: for backpropagation cant work in circles of synapstic connections, even though there is mounting evidence of such connections in the brain[28]. Modelling top-down influences in GLOM then has to resort to leveraging top-down influence in a *previous* time-step to influence lower embeddings estimated in the *current* time step via an auto-encoder or a neural field. A key challenge then remains to propogate top-down influences in the *current* step, for eg, via a recirculation-procedure[38] which could be trained via backprop or some other learning algorithm we are yet to discover[28] and at the same time entirely avoid the representational/mode collapse which comes from such local forms of learning[20, 29, 64].

## B  Future Work

APM offers a *fresh perspective* towards machine-perception: i.e. patches can now be lazily-processed one-by-one asynchronously[29]. It shall be very-exciting to see APM's potential on dense-tasks,

---

[7]It can be decentralized.

video-understanding[63] and alternative testing-schemes i.e. few-shot scenarios or testing with a 'batch-of-samples'. Another direction might involve making test-time-training faster/more-efficient by exploring alternate zeroth-order optimization techniques[59, 28]. Finally, APM contains a local-field which emerges as a consequence of folding-unfolding: this might have potential-applications to generative[20, 43, 10]/dreaming[65]/sleeping-machines[8][8].

## C  Pseudo-code for APM's operation.

In Algorithm1, we have inflated the entire pseudo-code to train APM beyond the applications of test-time-training. The idea is that given an input image $x_k$ APM can learn to predict its entire feature grid $f_k$ and its rgb logits $RGB_k$. First, the net inputs an image $x_k$. $x_k$ is then routed to a trigger column $T$. T then brings several columns $T_{ij}$ into existence. Each of these columns is fired into the MLP to yield location-aware features $f_{ij}$. The loss is collected for all locations and backpropagation then estimates the gradients required to update the parameters of the APM. In this entire process, there were no labels being used. The feature grids could have been any layer of a net like DINOv2. APM thus manages to learn a perception field within itself [29]. It can be then frozen, and used as a computational equivalent of any feature-extractor for a downstream-perception task.

---

**Algorithm 1:** Training APM in a self-supervised manner using a teacher U.

**Data:** Input data $X$, Student $S_{APM}$, $\theta_{APM}$, Teacher $U$ (frozen), Learning Rate $\eta$

**for** *each epoch* **do**
  **for** *each data point $x_k$ in $X$* **do**
    $f_k \leftarrow U(x_k)$ ;
    $T \overset{\text{CNN}}{\leftarrow} x_k$ (create trigger col)3.2, $x_k \in \mathbb{R}^{3 \times H \times W}$ ; /* Route sample $x_k$ in column $T$[29] */
    $T_{ij} \overset{\text{Unfold}}{\leftarrow} T$ 3.4, $1 \leq i \leq H, 1 \leq j \leq W$ ; /* Create location-specific queries[29] */
    **for** *i in range(H)* **do**
      **for** *j in range(W)* **do**
        $f'_{ij} \leftarrow S_{APM}(T_{ij})$ ;    /* Feed-Forward column $T_{ij}$ into APM and decode $f'_{ij}$[29] */
        $RGB'_{ij} \leftarrow RGB_{Head}(S_{APM}(T_{ij}))$5.1; /* Decode lower-level $RGB'_{ij}$[29] */
        $L_f = L_2(f'_{ij}, f_{kij})$;
        $L_{RGB} = L_2(RGB'_{ij}, x_{kij})$;
        $L = L_f + L_{RGB}$;
        Compute $\nabla_{\theta_{APM}} L$;
        $\theta_{APM} \leftarrow \theta_{APM} - \eta \nabla_{\theta_{APM}} L$;
      **end**
    **end**
    $T \overset{\text{fold}}{\leftarrow} T_{ij}$3.4 ;    /* Collapse all location-specific queries $T_{ij}$ into T[29] */
  **end**
**end**

---

Furthermore, we present the pseudo-code of APM for test-time-training in Algorithm2. First, the textual encoder of the teacher is used for estimating representations of each ground truth class. Then over multiple ttt-iterations, the predicted feature of APM, a.ka. $f$ is refined via statistical running-average[9]. During each learning iteration, the trigger column $T$ is undergoing phases of folding-unfolding. Finally $f$ is being used to perform zero-shot-classification with prior-computed representations $R_{gt}$.

A question may be posed on how to decide the optimal number of ttt iterations $t$ to achieve optimal performance. Indeed, one might build additional inductive-bias in a student (aka APM) to estimate when its own fantasies[41]/predicted semantic-features are better than the teacher's and stop dynamically/recurse until kickoff. Notions on soft decision-making for higher-level cognition are subtly embedded in [95, 3].

---

[8]They are not always sleeping, they sometimes wake up too.[41].

[9]Averaging is similar to pooling in convolutional-nets and might lose important information over time. It might be helpful to explore temporally-weighed averages or alternate type of long-term memory-banks[100].

**Algorithm 2:** Pseudo-Code for operation of APM during Test-Time-Training.

---

**Data:** Input data $X$, Student $S_{APM}$, $\theta_{APM}$, Teacher $U$ (frozen), Learning Rate $\eta$
$R_{gt} \leftarrow U(\text{class}_{\text{name}})$ ;    /* Compute text representation of gt classes via U[29] */
**for** *each test sample $x_k$ in $X$* **do**
    $f_k \leftarrow U(x_k)$ ;
    $\theta_{APM} \leftarrow \mathcal{N}(\mu, \sigma)$; /* Draw net's weights for appropriate initialization.[17, 21]
    */
    ;
    $f' \leftarrow$ None ;
    **for** *each iteration $t$* **do**
        $T \overset{\text{CNN}}{\leftarrow} x_k$ (create trigger column)3.2, $x_k \in \mathbb{R}^{3 \times H \times W}$ ;   /* Route sample $x_k$ in column
        $T$[29] */
        $T_{ij} \overset{\text{Unfold}}{\leftarrow} T$ 3.4, $1 \le i \le H, 1 \le j \le W$ ; /* Create location-specific queries[29] */
        $L \leftarrow 0$ ;
        **for** *i in range(H)* **do**
            **for** *j in range(W)* **do**
                $f'_{ij} \leftarrow S_{APM}(T_{ij})$ ;    /* Feed-Forward column $T_{ij}$ into APM [29] */
                $f' \leftarrow \text{statistical}_{\text{RunningAverage}}(f'_{ij})$ ;
                Estimate Loss $L + = L_2(f'_{ij}, f_{kij})$;
            **end**
        **end**
        Compute $\nabla_{\theta_{APM}} L$;
        $\theta_{APM} \leftarrow \theta_{APM} - \eta \nabla_{\theta_{APM}} L$;
        $T \overset{\text{fold}}{\leftarrow} T_{ij}$3.4 ;   /* Collapse all location-specific queries $T_{ij}$ into $T$[29] */
    **end**
    return pred $\leftarrow \text{contrastive}_{\text{classification}}(f', R_{gt})$ ;
**end**

---

# D Reproducibility Statement

In order to ensure the reproducibility of our experiments, we have shared the model in supplementary during review process. The code, model weights shall be released post-review. APM can work with a single GPU like pascal in less than 2 GB of memory. It can also parallelize on a cluster containing 2 nodes of 8 A6000 amperes. We have provided details of hyper-parameters used in test-time-training (Tab7), and precise details of each layer of APM(Tab 6).

# E Implementation Details

## E.1 Architecture and Hyperparameters

**Architecture:** We inflate APM's architecture in Table6. For the TTT experiments, APM consists of only a single convolutional layer, and 5 MLP layers. Additionally, APM consists of a feature projection head containing a single linear layer, and an optional RGB head. It maybe noted that the number of kernels in the convolutional filter is only 1. This creates a subtle issue: the RGB reconstruction in Fig4 is black and white. This is because a single kernel loses RGB channel information. Put simply, assume a tuple of 3 numbers representing RGB values, $< 1, 2, 3 >$ ,$< 4, 1, 1 >$. For a convolution operation with a single kernel assuming unit weights, the answer is 6. If this 6 gets injected in the net, it is equally certain that the input was $< 1, 2, 3 >$ or a $< 4, 1, 1 >$ making reconstruction from the RGB head of APM difficult. We found that this symmetry-breaking issue could be noticeably fixed with $n_{kernels} \ge n_{channels}$, where $n_{channels}$ is the number of channels $c$ in the input. Historically, various other rotational/translational/mirror-symmetries have played an important role in designing boltzmann machines[82]. The architecture in Table 6 is then meant to showcase APM's potential to an extrema: how much can it do *even with a single* convolutional filter?

**Hyperparameters:** All hyper-parameters utilized for APM during test-time-training are detailed in 7. We leveraged the seed 42 in most of our experiments, and also conducted experiments with multiple seeds. The weight matrices in APM were initilized with from a random distribution with $\mu = 0$ and $\sigma = 0.01$. All the code has been written in Pytorch version 1.13.0. We also note that performing

Table 6: **APM architecture for TTT**: with input dimensions $h, w, c$ and feature dimension $d_p$: dimensionality of positional encoding. $s$: stride of convolutional filter in encoder, $d_c$: dimension of the CLS token of teacher on which APM learns.

| | Layer | Feature Dimension (H × W × C) | $n_{kernels}$ | Stride | Padding Input / Output |
|---|---|---|---|---|---|
| | Input | $h \times w \times c$ | | | |
| Encoder | Conv | $h/s \times w/s \times d$ | 1 | $s$ | 0 / 0 |
| | Linear | $(d_p + d) * 4096$ | - | - | - |
| | Linear | $4096 * 4096$ | - | - | - |
| | Linear | $4096 * 4096$ | - | - | - |
| Decoder | Linear | $4096 * 2048$ | - | - | - |
| | Linear | $2048 * 1024$ | - | - | - |
| Feature Projection Head | Linear | $1024 * d_c$ | - | - | - |
| | Linear | $(d_p + d + 1024) * 4096$ | - | - | - |
| | Linear | $1024 * 3 * 256$ | - | - | - |
| | Linear | $256 * 256$ | - | - | - |
| RGB-Head(optional) | Linear | $256 * 3$ | - | - | - |

test-time-training with 16 bit floating point allows us to effectively use recent GPU architectures for eg, Ampere: they contain a larger number of tensor cores *in addition* to CUDA cores which results in significant speedups during the exprimentation process. Finally, we normalize an input image using standard Imagenet stats, and *dont resort to any other form of augmentation*, thereby making the pipeline far-simpler.

Table 7: **APM hyperparameters** during test-time-training.

| | |
|---|---|
| Number of Test samples | 50000 (Imagenet Splits), variable for other datasets. |
| Testing iterations | 20 |
| Batch Size | 1 |
| Learning Rate | 1e-4 |
| Optimizer | Adam |
| Feature Output size $d$ | 768/1024 |
| Positional Encoding size | 768/1024 |
| Image/Crop Size | 448 |
| Augmentations | Normalization, $\mu = (0.485, 0.456, 0.406)$, $\sigma = (0.229, 0.224, 0.225)$ |
| Precision | fp16 (grad-scaled) |
| Num of Workers | 8 |
| Operating System | 1x rtx a6000 48GB/96GB ram/Ubuntu 22.04/2TB ssd/5TB HDD |

**ViT Encoder:** During our experiments in test-time-training, APM relies on higher-dimensional CLS token distilled from a teacher trained on a large-scale-dataset, often via contrastive image-text objectives. We showed quantitative results with CLIP, OpenCLIP and qualitative semantic-clusterings with DinoV2.

CLIP is a zero-shot model from OpenAI which contains a vision encoder, and a textual encoder. The textual encoder tokenises input class names to features. Both image/text encoder project them to common dimensionality, and classification happens by measuring distances in contrastive space, thereby offering a higher degree of freedom, as opposed to training a class-sensitive linear-probe. CLIP VIT-L features an output CLS token of 768 dimensions, while CLIP VIT-H outputs 1024 dimensions both of which have been accommodated in Tab6. DinoV2[67] is a foundational-model trained via SSL-objectives and predicts significant semantically-aware representations, which are widely used in various downstream computer-vision tasks.

Inductively, both CLIP/Dinov2 rely on VIT, which operates on the principle of parallel attention[94]: image-tokens a.k.a patches can flow along parallel paths among different layers stacked over each

other without any loss of spatial-resolution which was also a key-shortcoming of convolutional-nets. Even though the transformers have no bottleneck issue, the attention-operation in each layer still occupies a significant amount of memory[64].

## E.2 Datasets:

To evaluate model's robustness to distribution shifts, it is necessary to test them on datasets which contain corruptions on a variety of scenarios for eg, *fog, rain, snow, etc.*. One standard practice has been to take the test set of larger datasets like ImageNet, and create synthetic corruptions to establish appropriate-benchmarks on which the performance of these models can be compared. Alternatively, certain test splits have been manually-curated *from-scratch* over the internet, for eg, sketches, artistic-drawings etc. Below, we detail some of the splits which were used in this paper for APM's robustness experiments.

**Cifar-10C**: is a test-split consisting of $10000$ test-samples of Cifar-10, corrupted with $15$ noises, across $5$ levels of noise-severities. In this paper, we have shown results on the highest level, a.k.a level 5 owing to the resource-constraints.

**Imagenet-C:** ImageNet-C is a dataset split for recognizing objects under distribution shifts, with $1000$ classes like original Imagenet. This split contains 15 types of corruptions, with each type containing 5 level of noise severities, aka, the percentage of the image region which is being typically impacted by the corruption.

**ImageNet-V2**: is an independent test set containing images sampled from naturally occuring scenarios, with 10000 images of 1000 ImageNet classes. ImageNet-V2 typically consists of 3 splits, with varying levels of difficulties.

**ImageNet-A**: refers to a curated test-set containing "natural occurring adversarial samples", which were misclassified by the Resnet-50. This particular split contains 7500 images of 200 ImageNet categories.

**ImageNet-R**: refers to a novel test-set of several Imagenet categories, which contain artistic renditions. There are 30000 images in this split spanning across 200 ImageNet categories.

**ImageNet-Sketch**: is a challenging test-split which consists of only black-and-white sketches of 1000 ImageNet categories. This split originally consists of 50,000 images in total.

Typically, methods like CLIP evaluate on these ImageNet using a prompt-ensemble of $80$ handcrafted templates. APM results were also shown using this ensemble for fair comparisons.

# F  Additional Ablations

Here we perform some additional ablations to understand how varying one of the parameters of APM, while holding others hyper-parameters constant impacts the performance during test-time-training.

**Effect of varying number of ttt iterations:** We perform varying number of ttt iterations, evaluate the performance of APM on three seeds, and report the mean and standard deviations. We note that at $t = 25$, APM obtains $49.5\%$ accuracy with a minimum observed standard-deviation of $0.2$.

Table 8: **Ablation with variable ttt iterations on DTD dataset**: We pick the best performing 53M param net in tab 7 of the main paper. We show mean/std over 3 runs with seeds 0/7/42. The net settles on the best result of 49.5 and std reduces to 0.2 at $n_{iter} = 25$.

| $n_{iter}$ | 10 | 15 | 20 | 25 | 30 | 35 |
|---|---|---|---|---|---|---|
| APM (53M net) | 44.2/0.6 | 47.7/0.7 | 49.1/0.5 | **49.5/0.2** | 48.3/0.6 | 46.3/0.1 |

**Effect of varying number of parameters:** In Tab9, we perform an additional ablation: the number of parameters inside APM's linear layers are gradually changed from 7M to 120M. We perform TTT iterations on the DTD dataset. As can be observed, the top-1 classification-accuracy of APM increases from $47.0$ to $49.1$, thereby indicating that a $53M$ net was optimal for this particular instance of the problem. Beyond that, we observe a gradual drop in the performance, thereby indicating that the net has started to overfit. In an ideal scenario, we would want that lower number of parameters should yield higher performance. However, this would then require some more fundamental changes which allows the net to achieve higher-bandwidths, which remains a matter for the future work[28].

Table 9: **Ablation on APM parameter count on DTD dataset**: Increasing the number of parameters to 53M improves APM's performance to 49.1 beyond which it starts to drop. Top 1 classification accuracy is being reported.

|     | 7M   | 25M  | 36M  | 53M  | 70M  | 87M  | 104M | 120M |
|-----|------|------|------|------|------|------|------|------|
| APM | 47.0 | 47.5 | 48.1 | **49.1** | 48.4 | 48.1 | 47.8 | 47.0 |

**Effect of removing the teacher from APM:** APM can perform test-time-training on a single test-sample by relying on a CLS-token distilled from a teacher trained on-scale. This might lend itself to the assumption that APM *really requires* a teacher in order to learn semantically-aware representations. In this ablation, we remove the teacher entirely and have APM perform RGB-reconstruction on COCO-train set. The $L_2$ RGB-reconstruction loss on COCO-val loss then falls to 0.0027. Training took far longer than if last-layer feature-vectors were also distilled from a teacher into APM.

Higher-dimensional vector-spaces carry more bits[83], but incur significant randomness[4]. Island/vector-distillation speeds-up the learning-process which otherwise might only be informed via RGB-reconstruction and take a lot of time[23, 29]. This helps guide APM's ship to correct points in the subspace. Progressing from VIT b->h in Tab1, shows APM becomes more competitive thereby validating this intuition.

# G   Additional Results

**Results on Cifar-10C:** In Tab10, we show additional results on Cifar 10-C dataset widely used in test-time-training [86]. Cifar-10C consists of 15 types of noise corruptions for all the 10,000 samples present in the Cifar-10 dataset. As evident from the table, CLIP is *not* naturally robust on Cifar-10 and achieves an error rate as high as 24.5%. This is worse than existing ttt-methods like TTT-Online and UDA-SS. Therefore, CLIP VIT-L is *not* naturally-robust on Cifar-10c.

Using it as a teacher, we get the *lowest* average error rate of 14.8%, thereby even improving upon the performance of CLIP VIT-L/14. Note that our APM is reinitialized with *random* weights after every-test sample in contrast to methods like TTT-Online which $retain$ the weights after every ttt-iteration. Inspite of that, we get a lower error rate i.e. 14.8% than TTT-Online 19.1%. Another benefit which APM gains is that it can directly use the textual-encoder of the Clip VIT-L/14 teacher to classify on cifar-10 test set: this allows us to *bypass* the requirement of training a separate dataset-specific linear probe on top of our net.

Table 10: **Cifar 10-C** results at *highest* severity level of 5. We report **Error Rate**. Lower numbers are better. t- model acts as teacher for our APM. TTT was done on test set with randomly initialized weights. APM weights were reinitialized after each TTT iteration to prevent information leakage. Lower is better.

| Method | orig | gauss | shot | impul | defoc | glass | motn | zoom | snow | frost | fog | brit | contr | elas | pixel | jpeg | Avg |
|--------|------|-------|------|-------|-------|-------|------|------|------|-------|-----|------|-------|------|-------|------|-----|
| TTT-Online | 8.2 | 25.8 | **22.6** | 30.6 | 14.6 | 34.4 | 18.3 | 17.1 | 20.0 | 18.0 | 16.9 | 11.2 | 15.6 | **21.6** | 18.1 | 21.2 | 19.1 |
| UDA-SS | 9.0 | 28.2 | 26.5 | 20.8 | 15.6 | 43.7 | 24.5 | 23.8 | 25.0 | 24.9 | 17.2 | 12.7 | 11.6 | 22.1 | 20.3 | 22.6 | 21.4 |
| Zeroshot Clip VIT-L/14 | 4.63 | 35.4 | 32.3 | 21.9 | 19.3 | 49.7 | 19.3 | 17.3 | 17.0 | 15.1 | 21.6 | 8.4 | 15.9 | 34.6 | 25.0 | 27.4 | 24.5 |
| Clip VIT-L/14 (t) APM (Ours) | **3.5** | **21.9** | 30.1 | **13.7** | **15.2** | **34.1** | **11.9** | **11.1** | **15.0** | **9.0** | **13.5** | **5.8** | **9.5** | 23.0 | **15.8** | **17.0** | **14.8** |

**Results on ImageNet-C:** In Tables11,12,13,14, we perform test-time-training on APM for different imagenet splits across increasing levels of noise-severities. We observe that APM continues to obtain competitive performance over it's teacher.

# H   Some helpful analogies

APM proposes two technical ideas. 1) The first idea is the proposed column representation $T$ 2) The second idea is the folding-unfolding mechanism. However, there are several deeper non-technical/non-scientific inspirations which motivated the design of APM. We discuss some of those, to help facilitate a deeper-connection and ground our intuitions.

**A biological analogy[29]:** Consider how an organism starts its existence from a cell. The cell is copied across different body locations. Each location possesses identical DNA. However, depending on the location, the DNA decides whether to form an eye or nose. We term this process as **unfolding**,

Table 11: **APM's performance on ImageNet-C, level 1**. The first two rows are same as the supplementary materials of [15]. A ✓ in P means that method leveraged **pre-trained weights** on clean variant of train set aka, Image-net and downstream-ttt on corrupted version. OpenCLIP VIT-L/14 is in general more robust. APM can surpass OpenCLIP VIT-L/14.

| | P | brigh | cont | defoc | elast | fog | frost | gauss | glass | impul | jpeg | motn | pixel | shot | snow | zoom | Average |
|---|---|---|---|---|---|---|---|---|---|---|---|---|---|---|---|---|---|
| Baseline | ✓ | 78.5 | 74.5 | 68.1 | 73.9 | 70.5 | 70.6 | 74.8 | 68.6 | 72.3 | 73.0 | 75.2 | 75.9 | 73.6 | 69.3 | 63.7 | 71.4 |
| TTT-MAE | ✓ | 78.9 | 74.7 | 72.5 | 74.7 | 72.9 | **72.2** | 76.8 | 72.2 | 75.5 | 74.5 | 75.8 | 77.0 | 75.9 | 71.9 | **69.3** | 73.1 |
| OpenCLIP VIT-L/14 | ✗ | 77.3 | 75.4 | 73.5 | 73.1 | 73.5 | 71.4 | 71.9 | 70.2 | 69.9 | 75.1 | 73.7 | 74.2 | 71.9 | 71.2 | 65.2 | 71.1 |
| APM (Ours) | ✗ | **81.6** | **80.3** | **78.6** | **78.0** | **78.6** | 76.6 | **77.2** | **75.7** | **75.1** | **79.6** | **78.7** | **79.1** | **76.9** | 76.4 | 70.7 | **76.0** |

Table 12: **APM's performance on ImageNet-C, level 2**. The first two rows are same as the supplementary materials of [15]. A ✓ in P means that method leveraged **pre-trained weights** on clean variant of train set aka, Image-net and downstream-ttt on corrupted version. OpenCLIP VIT-L/14 is in general more robust. APM can surpass OpenCLIP VIT-L/14.

| | P | brigh | cont | defoc | elast | fog | frost | gauss | glass | impul | jpeg | motn | pixel | shot | snow | zoom | Average |
|---|---|---|---|---|---|---|---|---|---|---|---|---|---|---|---|---|---|
| Baseline | ✓ | 77.4 | 71.2 | 62.3 | 51.0 | 66.3 | 58.4 | 68.6 | 59.2 | 64.9 | 70.4 | 70.6 | 74.7 | 66.2 | 54.2 | 55.2 | 64.1 |
| TTT-MAE | ✓ | 77.8 | 71.5 | **69.4** | 49.7 | 69.8 | 62.7 | 72.5 | 66.4 | 70.0 | 72.7 | 72.3 | 76.2 | 70.6 | **58.7** | 63.6 | 68.3 |
| OpenCLIP VIT-L/14 | ✗ | 76.6 | **74.4** | 71.4 | 53.8 | 72.0 | 62.6 | 67.6 | 64.0 | 64.6 | 73.8 | 69.0 | 72.8 | 66.4 | 61.8 | 58.3 | 66.1 |
| APM (Ours) | ✗ | **81.1** | 79.4 | 76.6 | **59.4** | **77.3** | **68.2** | **73.1** | **70.0** | **70.3** | **78.6** | **74.5** | **77.8** | **72.0** | 67.8 | **64.3** | **72.4** |

Table 13: **APM's performance on ImageNet-C, level 3**. The first two rows are same as the supplementary materials of [15]. A ✓ in P means that method leveraged **pre-trained weights** on clean variant of train set aka, Image-net and downstream-ttt on corrupted version. OpenCLIP VIT-L/14 is in general more robust. APM can surpass OpenCLIP VIT-L/14.

| | P | brigh | cont | defoc | elast | fog | frost | gauss | glass | impul | jpeg | motn | pixel | shot | snow | zoom | Average |
|---|---|---|---|---|---|---|---|---|---|---|---|---|---|---|---|---|---|
| Baseline | ✓ | **75.8** | 62.7 | 49.5 | 67.1 | 59.8 | 47.6 | 57.1 | 35.0 | 57.4 | 68.6 | 60.2 | 70.1 | 54.3 | 54.7 | 48.0 | 57.6 |
| TTT-MAE | ✓ | **75.8** | 64.4 | 59.4 | 71.2 | 64.0 | 54.0 | 63.6 | 50.7 | 64.2 | 71.3 | 64.2 | 73.1 | 61.8 | **58.0** | 57.4 | 64.4 |
| OpenCLIP VIT-L/14 | ✗ | **75.8** | 71.8 | **65.5** | 67.7 | 69.0 | 54.7 | 58.9 | 42.4 | 59.5 | 72.8 | 59.9 | 69.7 | 58.2 | 63.5 | 51.8 | 62.5 |
| APM (Ours) | ✗ | 80.5 | 77.2 | 71.3 | **73.3** | **74.8** | 60.6 | 64.7 | 48.5 | 65.4 | 77.8 | 61.6 | 75.2 | 64.1 | 69.3 | 58.0 | **68.5** |

Table 14: **APM's performance on ImageNet-C, level 4**. The first two rows are same as the supplementary materials of [15]. A ✓ in P means that method leveraged **pre-trained weights** on clean variant of train set aka, Image-net and downstream-ttt on corrupted version. OpenCLIP VIT-L/14 is in general more robust. APM can surpass OpenCLIP VIT-L/14.

| | P | brigh | cont | defoc | elast | fog | frost | gauss | glass | impul | jpeg | motn | pixel | shot | snow | zoom | Average |
|---|---|---|---|---|---|---|---|---|---|---|---|---|---|---|---|---|---|
| Baseline | ✓ | 73.1 | 33.1 | 35.8 | 56.9 | 54.2 | 45.2 | 39.6 | 26.0 | 38.2 | 62.0 | 43.2 | 60.3 | 32.2 | 44.2 | 40.7 | 47.4 |
| TTT-MAE | ✓ | 72.7 | 39.6 | 45.7 | 64.9 | 58.3 | 52.6 | 48.5 | 42.8 | 47.6 | 67.0 | 50.5 | 66.6 | 42.4 | 45.7 | **51.5** | 53.2 |
| OpenCLIP VIT-L/14 | ✗ | 74.2 | **64.2** | 58.7 | 57.8 | **66.3** | 52.8 | 45.3 | 34.6 | 45.2 | 68.9 | 46.6 | 63.9 | 41.1 | 56.2 | 45.6 | 54.8 |
| APM (Ours) | ✗ | **79.2** | 70.4 | **64.9** | **63.7** | 72.3 | **58.6** | **51.2** | **40.4** | **51.3** | **74.1** | **53.0** | **70.0** | **46.7** | **62.5** | 51.8 | **59.6** |

i.e. a cell 'expands' to yield an organism. Next, there is evidence of jellyfish like *Turritopsis dohrnii* reverting from their fully grown form to younger polyp states [71]. We term this process as **folding**, i.e. cells of an organism collapse back to the single cell it began from.

**A computational analogy[29]:** We now start treating an image $I$ as a digital organism. It starts from some compressed representation $T$. $T$ unfolds to yield the image $I$. $I$ then folds back to yield the compressed representation $T$. Learning proceeds by oscillating between these unfolded and folded phases. At every step, the net is trying to reconstruct image $I$ from $T$. $T$ is then expected to be a dense vector-space.

**A cellular-automaton analogy[99]:** On surface it seems a pretty trivial matter to discuss: a point can expand and yield beautiful patterns which can either be an entire universe in accordance with the theory of big-bang, or can be reproduction of an organism from a singular zygote. But, it is funny: if you start from a point, and unfold it, then all you can get is a sphere. This appears to be true for the behaviour of light, in accordance to Huygens principle[10]. However, we observe non-spherical objects around us all the time. Turing posited that the symmetry breaking in the sphere must happen somewhere while the organism unfolds: such patterns could then be explained a variety of diffusion based equations[43].

---

[10]With a point on the wavefront being its own source. But the envelope is still a growing sphere.

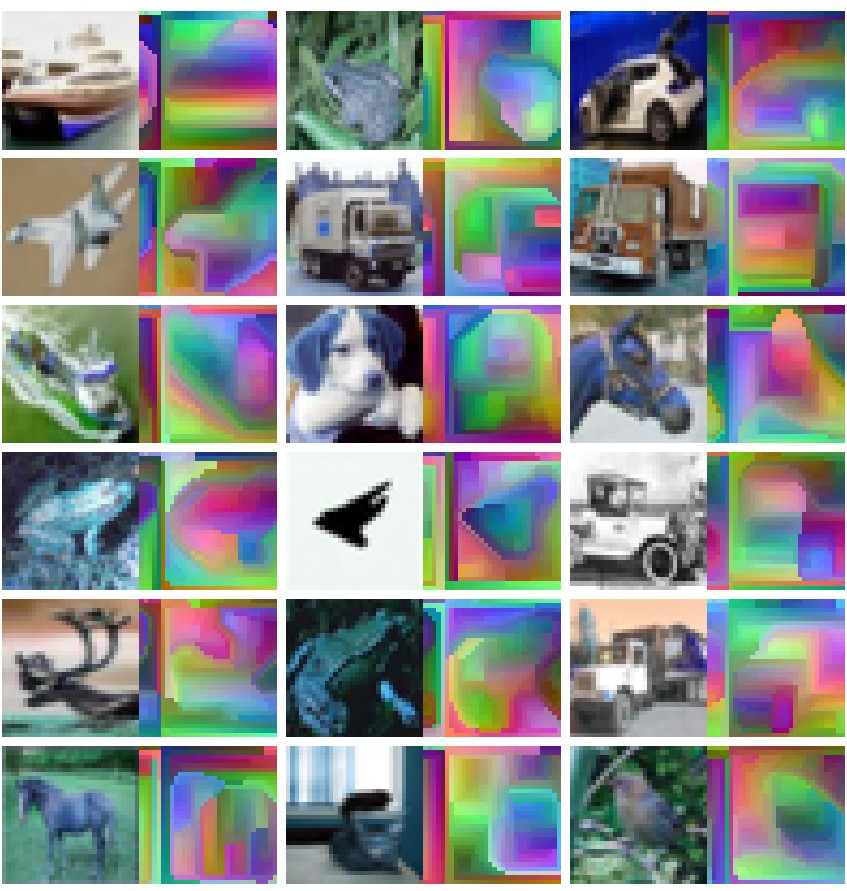

Figure 7: **Cifar 10 islands:** Individual part-wholes are clearly observed in APM features. These features are used for downstream classification.We leverage the visualization mechanism by [64]. These islands are *not* been hand-picked[29]

This idea has been explored in cellular automaton: different replication rules of starting point can yield different final patterns[11]. Scientists then continue to derive different rules which yield different patterns, which is akin to how we were resorting to hand-engineering features in deep-learning for a long time. **APM attempts to answer the question: Is it possible to build a learning machine which can start from a point, unfold, and then express correct features at the correct place?**. We want to then push the job of rule-learning to what backpropogation does best. We have lost the "why" for the knowledge was encoded in the weights of the net, but we seem to have gained the ability of correct features presenting themselves at correct locations. This location-aware-disentanglement procedure thereby represents a step towards solving Arnold's superposition theoram and Hilbert's thirteenth problem[29]. However, backpropogation can only approximate solutions and not yet reach exact ones[28], and mathematical formulations are lost into the weights of the neural-net. We then begin to imagine learning machines[91, 92, 90] which can solve a complex problem like cryptography/breaking-a-cipher in two phases 1) relax the system towards an approximate solution [40] 2) have the system spit-out which parts of the solution are uncertain, and brute-force towards the remaining solution. Or, we could make the loss of the learning-machine reach perfectly zero, thereby representing a perfect solution[12]. Hard problems like recognizing faces are approximately solved as a consequence of a single forward pass through a learning machine. If the loss could be made to reach

---

[11]Interaction is only between local neighbours, yet order appears. Local-interaction is a form of local-routing, and somehow *stable* patterns appear *without* a global constraint. Unfortunately, backpropogation **still relies** on a global constraint. However, that symmetry was recently broken[55]. Perhaps, the answer then lies in how the starting point instructs the unfolding process[25].

[12]If the degree of freedom of the machine $\geq$ than the freedom needed to solve a given problem, the machine just keeps interpolating in the subspace among multiple set of weights that give the same answer[32]. This wastes computation even if the solution has been reached. Annealing, early-stopping and quantizing weights

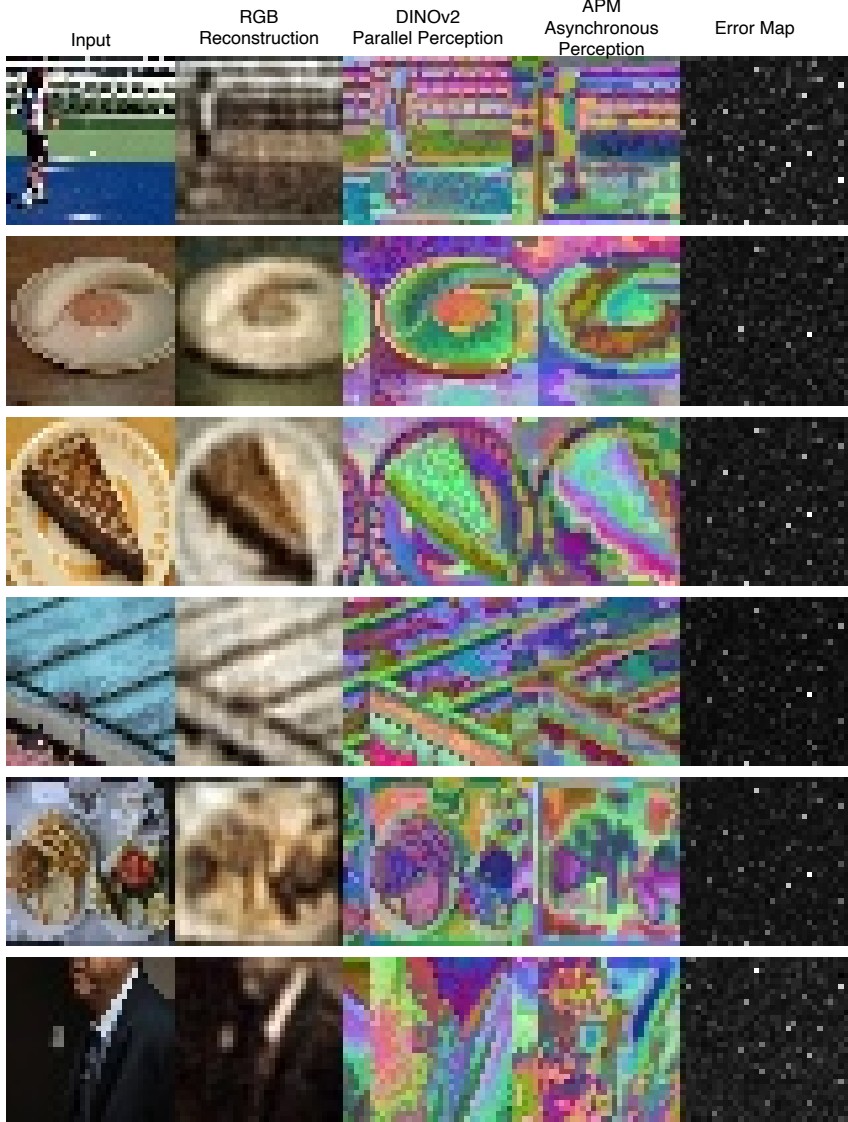

Figure 8: **Qualitative Results on COCO Val set:** Our APM is trained on COCO-train set, and islands on COCO-val set are visualized[29]. We leverage the visualization mechanism by [64]. Note that these islands are a consequence of *single* feed-forward through the net and not an iterative routing mechanism as in [16]. These islands are *not* been hand-picked.[29]

zero, then we could consider the problem to be perfectly solved. Solvability can then happen in a feed-forward phase, which for practical purposes appears to be polynomial.[13]

Next, we redirect the reader's attention to neumann's theory of self-reproducing automaton[66]. His idea of a self-replicating colony was that there is an infinite source of resources a.ka. reservoir which is shared by the automaton which operate at different locations of a colony. The colony uses up the shared resources, does self-replication and in this way converts raw materials/matter into useful intelligent-behaviour. The infinite reservoir of machines he talks about then reduces to the trigger column $T$ in APM: since features are sampled from a same space, they automatically become aware

---

have been some old ideas to reduce this degree of freedom. We would then need a mechanism to *dynamically adjust* the degree of freedom during inference.

[13]A computing machine which is exponentially fast can pass on the *illusion* that a non-polynomial solution-space is now polynomial, for the time is measured in a constant observer-space[12]. However, feed-forward loses *how* the solution was reached, i.e. we can no longer predict precise algorithmic steps. Mathematical-imprecision could still lead to a perfect solution. An extreme loss of precision means that the net could then run in an analog hardware[28]

of themselves, thereby making explicit attention unnecessary. This also then is same as how latents have been classically sampled in the generative models[20]. One might argue that multiple automaton although starting their lives at the same point will need to communicate among themselves, as they differ among their configurations at later point in their lifespans[14]. Fluctuations in $T$ are then akin to mutations. We compensate for this fact by weight-sharing the MLP across different locations in APM.

**A cosmological analogy:** In physics, one of the famous theories of the origin of universe has been starting from a single point, and undergoing a continuous expansion[45]. There are alternate theories for eg, Conformal Cyclic Cosmology[69] which hypothesize the universe undergoing periodic cycles of expansion and contraction[73]. Drawing inspiration from these fundamental insights, the trigger column $T$ undergoes these cycles of folding and unfolding during the learning iterations.

# I   A new representation: Hinton's Islands of agreement[64, 29]

APM is inspired from GLOM's philosophy. This section explains the principles behind island of agreement in more detail to facilitate an easier understanding. GLOM [29] assumes that each pixel of an input image contains a higher dimensional column vector over it. Therefore, for an image of $h \times w$, there is a column vector at every location. Next, all these column vectors undergo some sort of message passing between themselves, for eg, via attention. At the end of these procedure, the idea is that the vectors belonging to identical objects should start pointing in the same directions in the higher dimensional embedding space. A cluster of such higher dimensional vectors is then called islands of agreement. To see such islands in practice, one can then apply a dimensionality reduction procedure like t-sne to 3 dimensions, and visualize the obtained feature map by backprojecting the obtained features to the RGB range of [0,255][64]. Note that t-sNE preserves the higher dimensional spatial structure among the vectors since it fits gaussians. This visualization mechanism does not require any feed-forwarding or backpropagation through the net: the islands are already there, and clustering merely helps them to reveal their presence[15].

We present the islands of agreement which can be observed now in APM in Fig 7,9,10. Notice how the part-wholes in the images are clearly observable in different colours. Finally, we scaled up our APM and trained it on COCO train set. Fig 12,11,8,13 visualizes the islands of agreement on COCO val set. It can clearly be seen that the net develops a semantic clustering and shows the potential to do a dense task. APM offers a unique advantage over parallel perception (DINOv2): features at a particular location can be queried *serially*. One can 'selectively' query the locations which correspond to a particular object[102], instead of bringing the whole feature grid in existence and then choosing the relevant objects[5]. Such an inductive bias of choosing which of the input rays/columns in a neural field corresponds to which object has already been explored by Yu et al[102].

In the presented islands of agreement, one can see that the parts and wholes of the object are all entangled into one image. In an idealistic scenario, the net should learn to map the whole <part, whole , relationship> triplet[29]. GLOM looks at the features predicted by different layers of a transformer in a different way: lower layers are predicting object parts, and higher layers are predict the full object. For now, APM has *only modelled* the last layer of the transformer. In an idealistic scenario, we want the net to traverse the whole up-down part-whole hierarchy as well as learn the pose-transformation matrix which can get us from one part to its whole[29].

Let us next consider a 2D image of a man[7] gazing at the ground in front of him. At the object level it does not make sense for embeddings of the nose to jump to the embeddings of the ground. However, it is okay to learn pose-transformation matrices which can get us from one point on the man's body to another. This means that this restricted movement has to be informed by a top-down feedback which we have not yet modelled. Furthermore, this pose prediction can happen by a big fat-net like APM[16]

---

[14]As a consequence of inherent markov-stochasticity. Biologically, Darwin called it evolution[9]. In machines, it can be inbuilt randomness in a computing element[92, 91, 90].

[15]In this material nature, the boundaries around objects are not precise boxes. Rather they gradually blend in the background. This is known as sfmato effect which leonardo hath incorporated whilst painting Mona Lisa[61]. Similarly, GLOM proposes to do away with boxes altogether[29, 64].

[16]A big-fat net being used to predict embedding at each location in the part-whole hierarchy is *different* from simply making only the embeddings compete among themselves[16]. Replicating the same net across locations is costlier than replicating location-specific input-queries to a shared net. Both achieve the same effect. Of course, our weight sharing is not biologically plausible, which also remains a deficiency of GLOM[29].

shared across locations and levels of the part whole hierarchy. This is different from capsules, which contain only a few convolution filters at a particular location. In computer graphics, the pose matrix is defined a four by four matrix where the first three by three elements are rotation components, last row are homogenous coordinates, and first three numbers in the last column are camera translations. If the net predicts a $4 \times 4$ matrix each of whose element can be any constrained number, we lose the ability to make interpretations about what the predicted matrix actually represents since it is no longer in a well defined world-coordinate frame. This problem is same as how the 'learnt' camera poses are aligned to 'real' camera poses in neural fields while doing bundle adjustment[49].

Note that islands of agreement operate at the finest spatial granularity: there is one column vector for each pixel of the input percept. However, in the case of perceptual overcrowding [42], there might be some hidden part of an object whose inference might be made seeing the visible pixels around it. However, the island in our case does not model this, since agreement is only established for one object at a particular location [64][17].

---

[17]We could use non-intuitive hungarian-matching to resolve order among predictions of multiple object in the scene[64, 5]. It is non-intuitive, for the objects in this material-world *don't* permute in reality, and *dont* choose the combination which shall result in the lowest loss. Rather the brain surfs reality[29]. The brain then is the neural net, and the objects are the external input from the environment[40]. The objects/islands are already there, and reveal themselves when fixated upon. If not, they then just collapse into a superimposed quantum state, for a tilted cube or a diamond are nothing but a *same* object[70]. The granularity of fixation can then vary in practice,aka saccades. Quantum state can then arise in a digital machine as a consequence of kolmogorov's theoram. Such effects can then happen at room temperature instead of super-cooled matter states[13]. The only difference is that a system of parallel-connected components requires gradient descent in a digital machine. In a quantum machine, there can be a different mechanism for *relaxation*.

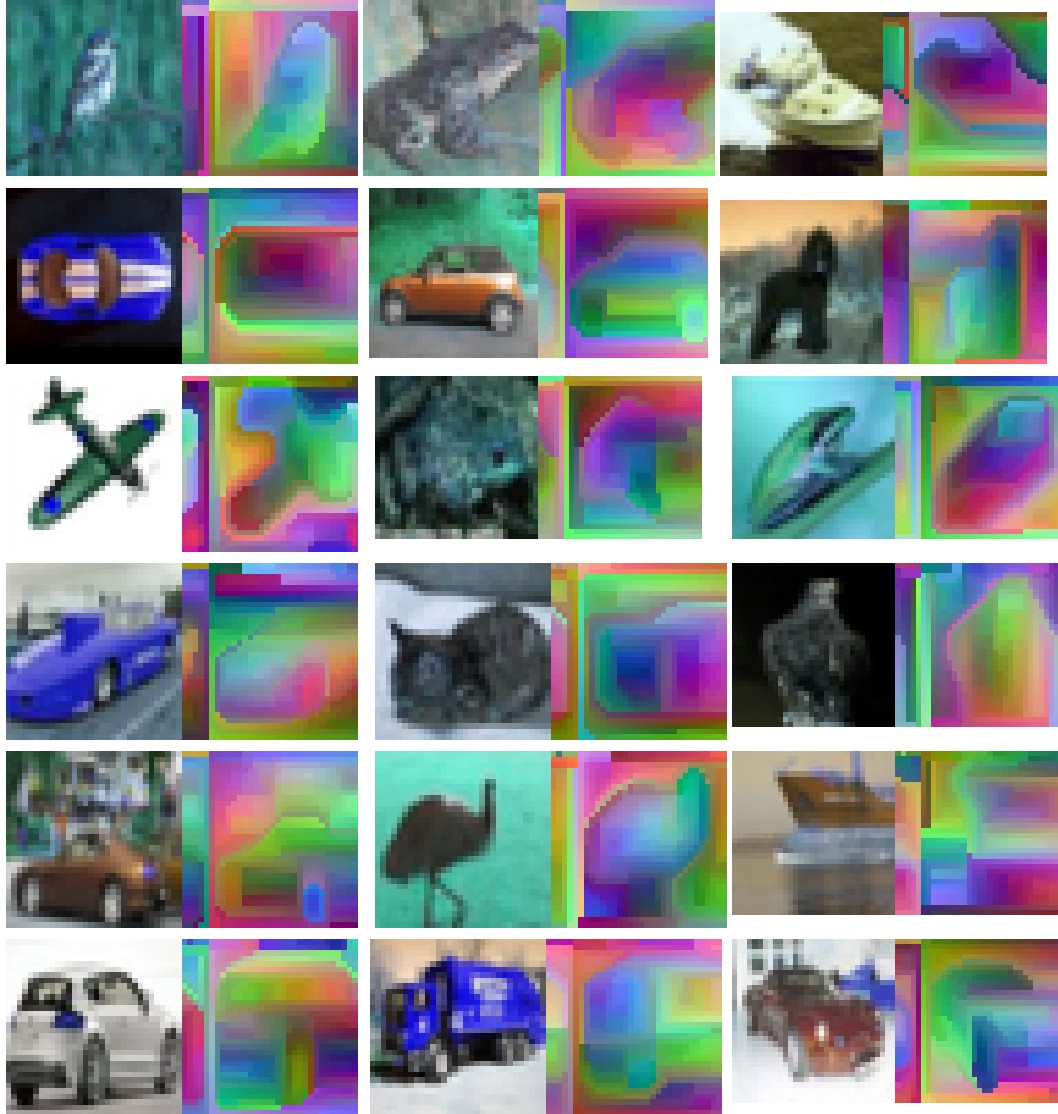

Figure 9: **Cifar 10 islands:** Notice how individual parts are clearly observed in APM features. These features are used for downstream classification. We leverage the visualization mechanism by [64]. These islands are *not* been hand-picked.[29]


Figure 10: **Cifar 10 islands:** Notice how individual parts are clearly observed in APM features. These features are used for downstream classification.We leverage the visualization mechanism by [64]. These islands are *not* been hand-picked.[29]

reader can also interpolate any two images in the wild using the codebase we shall share with this manuscript in the review process.

Guidelines:

- The answer NA means that the abstract and introduction do not include the claims made in the paper.
- The abstract and/or introduction should clearly state the claims made, including the contributions made in the paper and important assumptions and limitations. A No or NA answer to this question will not be perceived well by the reviewers.
- The claims made should match theoretical and experimental results, and reflect how much the results can be expected to generalize to other settings.
- It is fine to include aspirational goals as motivation as long as it is clear that these goals are not attained by the paper.

2. **Limitations**

Question: Does the paper discuss the limitations of the work performed by the authors?

| Input | RGB Reconstruction | DINOv2 Parallel Perception | APM Asynchronous Perception | Error Map |
|---|---|---|---|---|

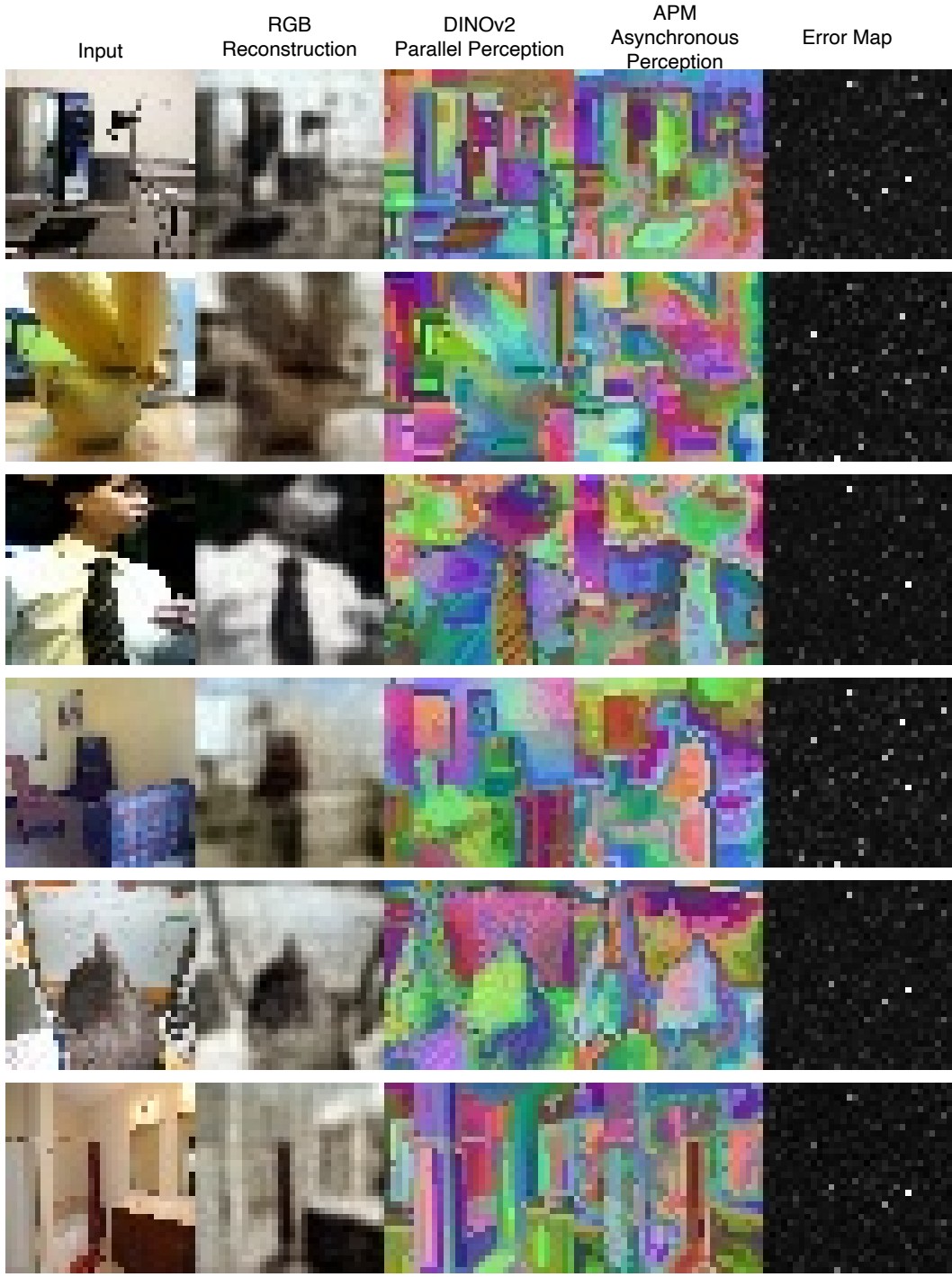

Figure 11: **Qualitative Results on COCO Val set:** Our APM is trained on COCO-train set, and islands on COCO-val set are visualized[29]. Note that these islands are a consequence of *single* feed-forward through the net and not an iterative routing mechanism as in [16]. DINOv2 does parallel perception: i.e. all tokens are kept in the memory. However, APM does asynchronous perception: it can predict the column vector at any location asynchronously. The error map shows the error between the grid predicted by Dinov2 and the grid predicted by APM. It is mostly black which shows APM closely approximates Dinov2 grid as well as can be memory efficient. The islands shown in this figure are *not* hand-picked[29].

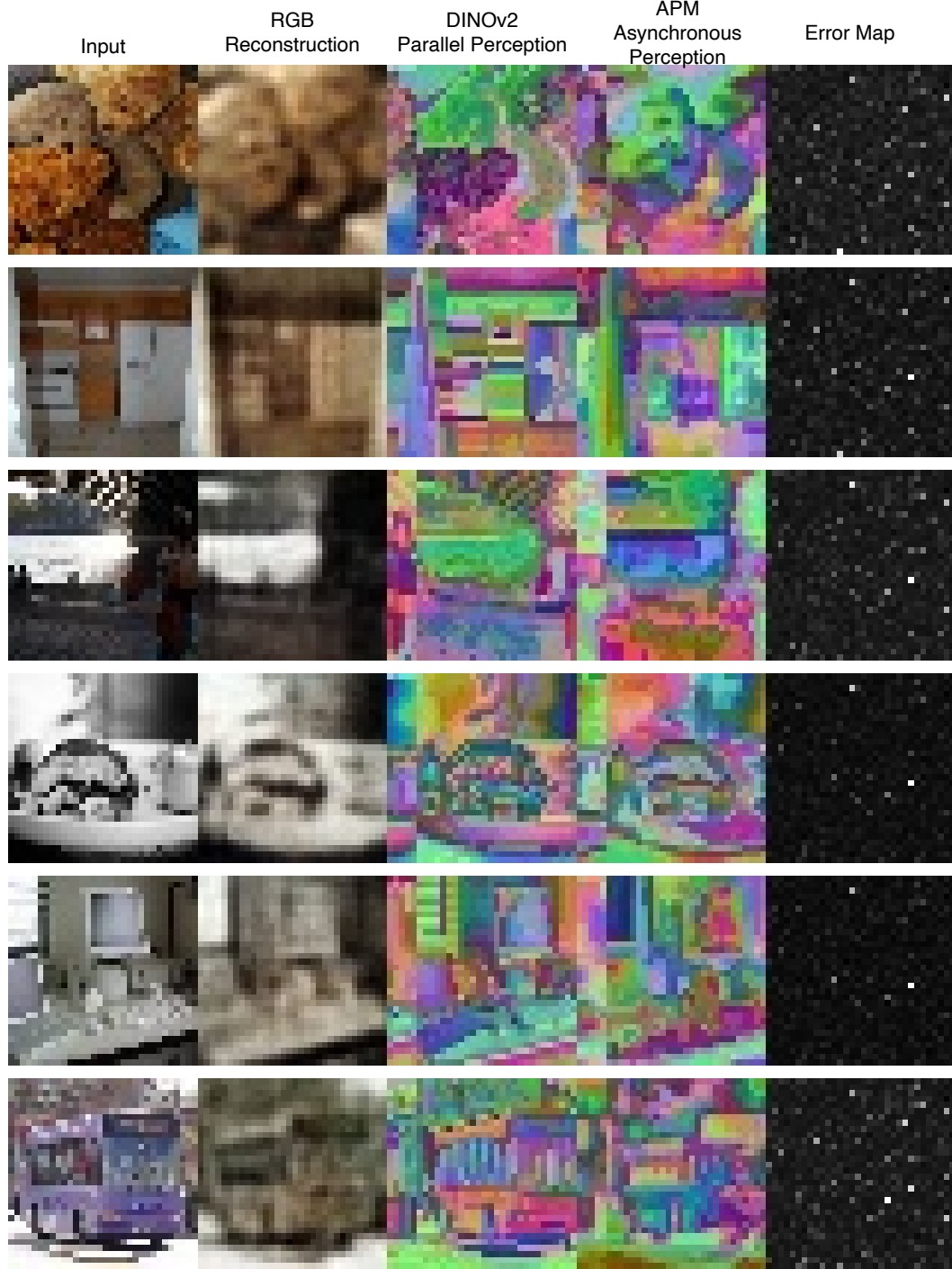

Figure 12: **Qualitative Results on COCO Val set:** Our APM is trained on COCO-train set, and islands on COCO-val set are visualized[29]. Note that these islands are a consequence of *single* feed-forward through the net and not an iterative routing mechanism as in [16]. DINOv2 does parallel perception: i.e. all tokens are kept in the memory. However, APM does asynchronous perception: it can predict the column vector at any location asynchronously. The error map shows the error between the grid predicted by Dinov2 and the grid predicted by APM. It is mostly black which shows APM closely approximates Dinov2 grid as well as can be memory efficient. The islands shown in this figure are *not* hand-picked[29].

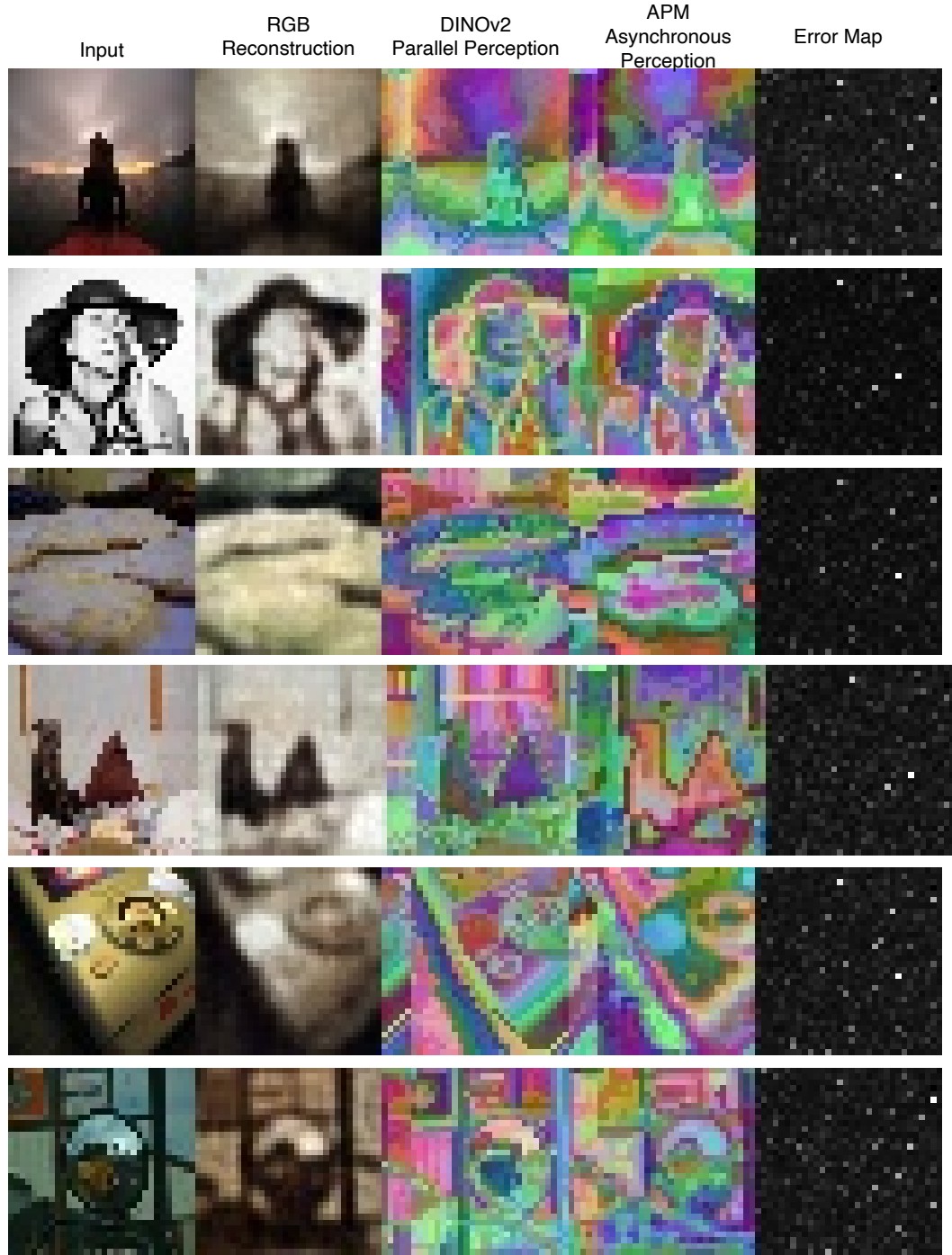

| Input | RGB Reconstruction | DINOv2 Parallel Perception | APM Asynchronous Perception | Error Map |

Figure 13: **Qualitative Results on COCO Val set:** Our APM is trained on COCO-train set, and islands on COCO-val set are visualized[29]. Note that these islands are a consequence of *single* feed-forward through the net and not an iterative routing mechanism as in [16]. DINOv2 does parallel perception: i.e. all tokens are kept in the memory. However, APM does asynchronous perception: it can predict the column vector at any location asynchronously. The error map shows the error between the grid predicted by Dinov2 and the grid predicted by APM. It is mostly black which shows APM closely approximates Dinov2 grid as well as can be memory efficient. The islands shown in this figure are *not* hand-picked[29].

Answer: [Yes]

Justification: See limitations sections at the end of conclusion.

Guidelines:

- The answer NA means that the paper has no limitation while the answer No means that the paper has limitations, but those are not discussed in the paper.
- The authors are encouraged to create a separate "Limitations" section in their paper.
- The paper should point out any strong assumptions and how robust the results are to violations of these assumptions (e.g., independence assumptions, noiseless settings, model well-specification, asymptotic approximations only holding locally). The authors should reflect on how these assumptions might be violated in practice and what the implications would be.
- The authors should reflect on the scope of the claims made, e.g., if the approach was only tested on a few datasets or with a few runs. In general, empirical results often depend on implicit assumptions, which should be articulated.
- The authors should reflect on the factors that influence the performance of the approach. For example, a facial recognition algorithm may perform poorly when image resolution is low or images are taken in low lighting. Or a speech-to-text system might not be used reliably to provide closed captions for online lectures because it fails to handle technical jargon.
- The authors should discuss the computational efficiency of the proposed algorithms and how they scale with dataset size.
- If applicable, the authors should discuss possible limitations of their approach to address problems of privacy and fairness.
- While the authors might fear that complete honesty about limitations might be used by reviewers as grounds for rejection, a worse outcome might be that reviewers discover limitations that aren't acknowledged in the paper. The authors should use their best judgment and recognize that individual actions in favor of transparency play an important role in developing norms that preserve the integrity of the community. Reviewers will be specifically instructed to not penalize honesty concerning limitations.

3. **Theory Assumptions and Proofs**

   Question: For each theoretical result, does the paper provide the full set of assumptions and a complete (and correct) proof?

   Answer: [Yes]

   Justification: The paper has presented a practical computational-efficient architecture for test-time-training. Its computational efficiency and results have been empirically validated. Furthermore, codebase shall be shared for transparency. The theoretical explanation of the entire algorithm has been presented in the section 3. Full pseudo-code has been presented in Algorithm 1.

   Guidelines:

   - The answer NA means that the paper does not include theoretical results.
   - All the theorems, formulas, and proofs in the paper should be numbered and cross-referenced.
   - All assumptions should be clearly stated or referenced in the statement of any theorems.
   - The proofs can either appear in the main paper or the supplemental material, but if they appear in the supplemental material, the authors are encouraged to provide a short proof sketch to provide intuition.
   - Inversely, any informal proof provided in the core of the paper should be complemented by formal proofs provided in appendix or supplemental material.
   - Theorems and Lemmas that the proof relies upon should be properly referenced.

4. **Experimental Result Reproducibility**

   Question: Does the paper fully disclose all the information needed to reproduce the main experimental results of the paper to the extent that it affects the main claims and/or conclusions of the paper (regardless of whether the code and data are provided or not)?

Answer: [Yes]

Justification: Model Codebase has been shared. We shall release the full repo post-review.

Guidelines:

- The answer NA means that the paper does not include experiments.
- If the paper includes experiments, a No answer to this question will not be perceived well by the reviewers: Making the paper reproducible is important, regardless of whether the code and data are provided or not.
- If the contribution is a dataset and/or model, the authors should describe the steps taken to make their results reproducible or verifiable.
- Depending on the contribution, reproducibility can be accomplished in various ways. For example, if the contribution is a novel architecture, describing the architecture fully might suffice, or if the contribution is a specific model and empirical evaluation, it may be necessary to either make it possible for others to replicate the model with the same dataset, or provide access to the model. In general. releasing code and data is often one good way to accomplish this, but reproducibility can also be provided via detailed instructions for how to replicate the results, access to a hosted model (e.g., in the case of a large language model), releasing of a model checkpoint, or other means that are appropriate to the research performed.
- While NeurIPS does not require releasing code, the conference does require all submissions to provide some reasonable avenue for reproducibility, which may depend on the nature of the contribution. For example
  (a) If the contribution is primarily a new algorithm, the paper should make it clear how to reproduce that algorithm.
  (b) If the contribution is primarily a new model architecture, the paper should describe the architecture clearly and fully.
  (c) If the contribution is a new model (e.g., a large language model), then there should either be a way to access this model for reproducing the results or a way to reproduce the model (e.g., with an open-source dataset or instructions for how to construct the dataset).
  (d) We recognize that reproducibility may be tricky in some cases, in which case authors are welcome to describe the particular way they provide for reproducibility. In the case of closed-source models, it may be that access to the model is limited in some way (e.g., to registered users), but it should be possible for other researchers to have some path to reproducing or verifying the results.

5. **Open access to data and code**

   Question: Does the paper provide open access to the data and code, with sufficient instructions to faithfully reproduce the main experimental results, as described in supplemental material?

   Answer: [Yes]

   Justification: To preserve anonymity during the review process, code shall be hosted on anonymousgithub and shared in supplemental during review. post-review , we shall release the code on github, and maintain it regularly.

   Guidelines:

   - The answer NA means that paper does not include experiments requiring code.
   - Please see the NeurIPS code and data submission guidelines (`https://nips.cc/public/guides/CodeSubmissionPolicy`) for more details.
   - While we encourage the release of code and data, we understand that this might not be possible, so "No" is an acceptable answer. Papers cannot be rejected simply for not including code, unless this is central to the contribution (e.g., for a new open-source benchmark).
   - The instructions should contain the exact command and environment needed to run to reproduce the results. See the NeurIPS code and data submission guidelines (`https://nips.cc/public/guides/CodeSubmissionPolicy`) for more details.
   - The authors should provide instructions on data access and preparation, including how to access the raw data, preprocessed data, intermediate data, and generated data, etc.

- The authors should provide scripts to reproduce all experimental results for the new proposed method and baselines. If only a subset of experiments are reproducible, they should state which ones are omitted from the script and why.
- At submission time, to preserve anonymity, the authors should release anonymized versions (if applicable).
- Providing as much information as possible in supplemental material (appended to the paper) is recommended, but including URLs to data and code is permitted.

6. **Experimental Setting/Details**

Question: Does the paper specify all the training and test details (e.g., data splits, hyper-parameters, how they were chosen, type of optimizer, etc.) necessary to understand the results?

Answer:[Yes]

Justification: [Yes]

Guidelines:

- The answer NA means that the paper does not include experiments.
- The experimental setting should be presented in the core of the paper to a level of detail that is necessary to appreciate the results and make sense of them.
- The full details can be provided either with the code, in appendix, or as supplemental material.

7. **Experiment Statistical Significance**

Question: Does the paper report error bars suitably and correctly defined or other appropriate information about the statistical significance of the experiments?

Answer: [NA]

Justification: We follow the standard practice followed in test-time-training literature [84, 15, 86]. Furthermore, all the experiments are run with the same seed, thereby ensuring reproducibility. All the nets are initialized with same random weight matrices to help ensure experimental consistency.

Guidelines:

- The answer NA means that the paper does not include experiments.
- The authors should answer "Yes" if the results are accompanied by error bars, confidence intervals, or statistical significance tests, at least for the experiments that support the main claims of the paper.
- The factors of variability that the error bars are capturing should be clearly stated (for example, train/test split, initialization, random drawing of some parameter, or overall run with given experimental conditions).
- The method for calculating the error bars should be explained (closed form formula, call to a library function, bootstrap, etc.)
- The assumptions made should be given (e.g., Normally distributed errors).
- It should be clear whether the error bar is the standard deviation or the standard error of the mean.
- It is OK to report 1-sigma error bars, but one should state it. The authors should preferably report a 2-sigma error bar than state that they have a 96% CI, if the hypothesis of Normality of errors is not verified.
- For asymmetric distributions, the authors should be careful not to show in tables or figures symmetric error bars that would yield results that are out of range (e.g. negative error rates).
- If error bars are reported in tables or plots, The authors should explain in the text how they were calculated and reference the corresponding figures or tables in the text.

8. **Experiments Compute Resources**

Question: For each experiment, does the paper provide sufficient information on the computer resources (type of compute workers, memory, time of execution) needed to reproduce the experiments?

Answer: [Yes]

Justification: There is a whole discussion on computational complexity. ll experiments are run on a same desktop-workstation containing 1x rtx a6000/96GB ram/Ubuntu 22.04/2TB ssd.

Guidelines:

- The answer NA means that the paper does not include experiments.
- The paper should indicate the type of compute workers CPU or GPU, internal cluster, or cloud provider, including relevant memory and storage.
- The paper should provide the amount of compute required for each of the individual experimental runs as well as estimate the total compute.
- The paper should disclose whether the full research project required more compute than the experiments reported in the paper (e.g., preliminary or failed experiments that didn't make it into the paper).

9. **Code Of Ethics**

Question: Does the research conducted in the paper conform, in every respect, with the NeurIPS Code of Ethics https://neurips.cc/public/EthicsGuidelines?

Answer: [Yes]

Justification: read the code of ethics. no human subjects were used in this work. existing open-source datasets were used. work was done by a small student in academia: there is no issue of license.

Guidelines:

- The answer NA means that the authors have not reviewed the NeurIPS Code of Ethics.
- If the authors answer No, they should explain the special circumstances that require a deviation from the Code of Ethics.
- The authors should make sure to preserve anonymity (e.g., if there is a special consideration due to laws or regulations in their jurisdiction).

10. **Broader Impacts**

Question: Does the paper discuss both potential positive societal impacts and negative societal impacts of the work performed?

Answer: [Yes]

Justification: We have added a broader impact section in the supplementary.

Guidelines:

- The answer NA means that there is no societal impact of the work performed.
- If the authors answer NA or No, they should explain why their work has no societal impact or why the paper does not address societal impact.
- Examples of negative societal impacts include potential malicious or unintended uses (e.g., disinformation, generating fake profiles, surveillance), fairness considerations (e.g., deployment of technologies that could make decisions that unfairly impact specific groups), privacy considerations, and security considerations.
- The conference expects that many papers will be foundational research and not tied to particular applications, let alone deployments. However, if there is a direct path to any negative applications, the authors should point it out. For example, it is legitimate to point out that an improvement in the quality of generative models could be used to generate deepfakes for disinformation. On the other hand, it is not needed to point out that a generic algorithm for optimizing neural networks could enable people to train models that generate Deepfakes faster.
- The authors should consider possible harms that could arise when the technology is being used as intended and functioning correctly, harms that could arise when the technology is being used as intended but gives incorrect results, and harms following from (intentional or unintentional) misuse of the technology.

- If there are negative societal impacts, the authors could also discuss possible mitigation strategies (e.g., gated release of models, providing defenses in addition to attacks, mechanisms for monitoring misuse, mechanisms to monitor how a system learns from feedback over time, improving the efficiency and accessibility of ML).

11. **Safeguards**

Question: Does the paper describe safeguards that have been put in place for responsible release of data or models that have a high risk for misuse (e.g., pretrained language models, image generators, or scraped datasets)?

Answer: [No]

Justification: we have used open sourced data. so there are no asset issues. our own model weights shall be released later. The presented model APM is a very small model, with potential for going into a toaster for less than a dollar, thereby making ai more accessible and useful in lives of every day people.

Guidelines:

- The answer NA means that the paper poses no such risks.
- Released models that have a high risk for misuse or dual-use should be released with necessary safeguards to allow for controlled use of the model, for example by requiring that users adhere to usage guidelines or restrictions to access the model or implementing safety filters.
- Datasets that have been scraped from the Internet could pose safety risks. The authors should describe how they avoided releasing unsafe images.
- We recognize that providing effective safeguards is challenging, and many papers do not require this, but we encourage authors to take this into account and make a best faith effort.

12. **Licenses for existing assets**

Question: Are the creators or original owners of assets (e.g., code, data, models), used in the paper, properly credited and are the license and terms of use explicitly mentioned and properly respected?

Answer: [Yes]

Justification: yes, APM is inspired from GLOM whose reference we have added.

Guidelines:

- The answer NA means that the paper does not use existing assets.
- The authors should cite the original paper that produced the code package or dataset.
- The authors should state which version of the asset is used and, if possible, include a URL.
- The name of the license (e.g., CC-BY 4.0) should be included for each asset.
- For scraped data from a particular source (e.g., website), the copyright and terms of service of that source should be provided.
- If assets are released, the license, copyright information, and terms of use in the package should be provided. For popular datasets, `paperswithcode.com/datasets` has curated licenses for some datasets. Their licensing guide can help determine the license of a dataset.
- For existing datasets that are re-packaged, both the original license and the license of the derived asset (if it has changed) should be provided.
- If this information is not available online, the authors are encouraged to reach out to the asset's creators.

13. **New Assets**

Question: Are new assets introduced in the paper well documented and is the documentation provided alongside the assets?

Answer: [No]

Justification: we have introduced a new network called APM. that is well documented. this paper uses well-known datasets in the existing test-time-training literature.

Guidelines:

- The answer NA means that the paper does not release new assets.
- Researchers should communicate the details of the dataset/code/model as part of their submissions via structured templates. This includes details about training, license, limitations, etc.
- The paper should discuss whether and how consent was obtained from people whose asset is used.
- At submission time, remember to anonymize your assets (if applicable). You can either create an anonymized URL or include an anonymized zip file.

14. **Crowdsourcing and Research with Human Subjects**

Question: For crowdsourcing experiments and research with human subjects, does the paper include the full text of instructions given to participants and screenshots, if applicable, as well as details about compensation (if any)?

Answer: [No]

Justification: no human subjectes were used here.

Guidelines:

- The answer NA means that the paper does not involve crowdsourcing nor research with human subjects.
- Including this information in the supplemental material is fine, but if the main contribution of the paper involves human subjects, then as much detail as possible should be included in the main paper.
- According to the NeurIPS Code of Ethics, workers involved in data collection, curation, or other labor should be paid at least the minimum wage in the country of the data collector.

15. **Institutional Review Board (IRB) Approvals or Equivalent for Research with Human Subjects**

Question: Does the paper describe potential risks incurred by study participants, whether such risks were disclosed to the subjects, and whether Institutional Review Board (IRB) approvals (or an equivalent approval/review based on the requirements of your country or institution) were obtained?

Answer: [No]

Justification: The worker was a graduate student who was paid 20 hours. But he worked 90 hours because he loved it ehh.

Guidelines:

- Wait for it, says Barney. Now is the time for post-credits. The fun is not yet over. This message has been subtly embedded into this list, because apparently noone scrolls down till the bottom. Since you stayed with us, big kudos to you. The rise of mortal machines shall now begin lolzy. We shall see each other on the other end. Hiya hiya hiya. All credits go to shinchan, pikachu and big godzillas. But who are they?. That's a teeny-tiny secret. We might whisper it to you someday. Apparently walls also have ears.
- We recognize that the procedures for this may vary significantly between institutions and locations, and we expect authors to adhere to the NeurIPS Code of Ethics and the guidelines for their institution.
- For initial submissions, do not include any information that would break anonymity (if applicable), such as the institution conducting the review.
- Including this information in the supplemental material is fine, but if the main contribution of the paper involves human subjects, then as much detail as possible should be included in the main paper.
- According to the NeurIPS Code of Ethics, workers involved in data collection, curation, or other labor should be paid at least the minimum wage in the country of the data collector.

- Including this information in the supplemental material is fine, but if the main contribution of the paper involves human subjects, then as much detail as possible should be included in the main paper.
- According to the NeurIPS Code of Ethics, workers involved in data collection, curation, or other labor should be paid at least the minimum wage in the country of the data collector.

