# OpenReview forum: "Asynchronous Perception Machine for Efficient  Test Time Training"
_NeurIPS.cc/2024/Conference — NeurIPS 2024 poster_

### Official Review · Reviewer_PHih · 2024-07-04

**Soundness:** 2
**Presentation:** 2
**Contribution:** 2
**Rating:** 6
**Confidence:** 4

**Summary:**

The authors propose a novel test-time training method called the Asynchronous Perception Machine. This method leverages knowledge distillation from other pretrained networks, such as CLIP. The main contributions of this approach are its robust accuracy in classifying corrupted images and its computational efficiency, requiring fewer FLOPs compared to other models.

**Strengths:**

This paper presents a creative and novel neural network architecture, demonstrating strong empirical results.

**Weaknesses:**

1. Lack of Related Works: The paper proposes a novel test-time training method but references fewer than five related works. However, there are more than 50 papers on test-time adaptation methods published in recent years (e.g., [1-8]), none of which are mentioned.
2. Lack of Comparison to State-of-the-Art Methods: The manuscript identifies TTT-MAE as the second-best method, yet this method was introduced in 2022. Since then, many methods, such as Diffusion-TTA [4] and DeYo [8], have surpassed TTT-MAE.
3. Insufficient Explanation of Proposed Method: The proposed method is not explained in sufficient detail. For instance, the Vision Encoder is only depicted in a figure and not discussed further in the text.
4. Misleading Information The “Trained” column in one of the tables is misleading. TTT-MAE does not require training data during the test phase; it requires an auxiliary loss during training.
5. Inadequate Ablation Study: The ablation study lacks relevance. While the authors propose adaptation to distribution shifts (i.e., corrupted images), the ablation is conducted only on CIFAR-10 and CIFAR-100, which are datasets of clean images.

Reference

[1] Tent: Fully Test-time Adaptation by Entropy Minimization

[2] Continual Test-Time Domain Adaptation

[3] Robust Test-Time Adaptation in Dynamic Scenarios

[4] Diffusion-TTA: Test-time Adaptation of Discriminative Models via Generative Feedback

[5] NOTE: Robust Continual Test-time Adaptation Against Temporal Correlation

[6] Robust Test-Time Adaptation in Dynamic Scenarios

[7] SoTTA: Robust Test-Time Adaptation on Noisy Data Streams

[8] Entropy is not Enough for Test-Time Adaptation: From the Perspective of Disentangled Factors

**Questions:**

It is very unclear what component in the Fig 1 is. Specifically, what computations do they do?

**Limitations:**

There is no concerned limitation.

---

> ### Author Rebuttal · Authors · 2024-08-06
>
> We thank the reviewer for providing us the chance to improve our work. Please find our responses:
>
> ```On improving  related-work``` As reviewer KZWi points out, we will discuss more recent tta/prompting/source-free domain-adaptation approaches. We have noted to add differences w.r.t feature refinement/distillation: a.k.a $L_2$ mimicking feature grids vs boltzmann-temperature-matching logits.
>
> ```On Clarifications between ttt/tta, comparisons with more state-of-the arts``` We apologize for the confusion between test-time-training (TTT)/test-time-adaptation (TTA), and would like to clarify the experimental setting of  test-time-training and its connections to test-time-adapation works, relative novelty of apm, after checking the shared works. Thank you for sharing these works and we will add these works to related work too.
>
>  - Existing works in TTA train on a source dataset and then adapt them for testing. APM, which follows TTT, works without such source-training by just initializing with ```randomized weights``` for only 1 test sample. TTA adapts to the target distribution (using all test samples), whereas TTT adapts to each sample independently.
> - TTA methods use clean datasets (eg c10) during train, and adapt on corrupted versions(c10c), which might  limit its potential to deploy on some other dataset [R1,R2,R3,R5,R6,R7], because they might require training dataset-specific linear-probe.
> - Some tta methods update statistics on "batch of test-set". APM can work with weights randomly-initialized after each sample.
> - APM is computationally efficient, since it uses only 1 conv layer and 5 mlp layers(tab 2, fig4), while some models might use higher-parameterized network like a ViT, or a diffusion-model [R1].
>
> We observe that [R1] also shows performance for TTT and below we compare APM with [R1] on datasets where APM performed well. We note our baseline numbers match that of [R4]. APM presents competitive performance, on bigger datasets like Imagenet. It is important to note that APM is much more efficient than [R1] (1conv/3MLP vs. ViT+Diffusion model). We will add these and other comparisons to Tab 3. Thank you so much for pointing this out.
> |                                   | Dataset  |          |             |            |         |
> | --------------------------------- | -------- | -------- | ----------- | ---------- | ------- |
> |                                   | ImageNet | Aircraft | Food101 |
> |                                   |          |          |             |            |         |
> | CLIP VIT-B/16 Baseline (from [1]) | 62.3     | 22.8     |  88.1    |
> | Diffusion-TTA [1]                     | 63.8     | 24.6     | **88.8**    |
> |                                   |          |          |                  |
> | CLIP VIT-B/16 Baseline (consistent with [9])           | 66.7     | 23.7     |  83.7    |
> | Ours                              | **68.1**     | **29.7**        | 84.2    |
>
> | Diffusion-TTA | APM |
> | --- | --- |
> | 865M  | **25M** |
> We also note lower number parameters in APM (25M) when compared to Diffusion-TTA (865M).
>
> - In some cases, methods like [R9] do show source-free domain-adaptation, but on smaller datasets. In tab 1-2, we have shown results on larger datasets like imagenet-c without imagenet pretraining on APM, as also duly noted in line 169.
> - We will be grateful to mention that APM is a fresh-perspective on machine perception, with the ability to do location-based processing.
>
> ```Details on vision-encoder``` We apologize for this confusion. It refers to the ViT/Dinov2 vision encoder. We will add the explanation in the paper.
>
> ```On misleading trained column in tables, confusion on ttt-mae``` We apologize for the confusion caused by this. TTT relies on two phases 1) train model with ssl task + labels on  train set. 2) test time -> use ssl task only. For eg., ttt-mae is initialized with pretrained imagenet-1k weights (sec 4.1 of their  paper) and needs ssl objective (masking) during ttt. However, APM is initialized with random weights, and no explicit ssl-task is required. We have renamed ```trained``` column to $P$ in all tables and clarified in caption: ```A ✓ in P means that method leveraged pre-trained weights on a clean variant of train set aka, Image-net and downstream-ttt on corrupted version```. We provide more results in global rebuttal pdf with updated captions. We will update tables in the main paper also.
>
> ```On Adding ttt-specific ablations``` We will be grateful to present this ablation with a variable number of ttt iterations on dtd below. Next, ttt with varying apm parameters is presented in tab 7 of paper. Further, apm's property of one-sample-overfitting during ttt (line 250), was evaluated in tab 5a).
> | n_iter | 10	| 15   | 20   | 25   	| 30   | 35   |
> |--------|------|------|------|----------|------|------|
> |    	|  	|  	|  	|      	|  	|  	|
> | 53M parameter  net (acc)   | 44.2/0.6 | 47.7/0.7 | 49.1/0.5 | **49.5/0.2** | 48.3/0.6 | 46.3/0.1 |
>
>
> At 25th iteration, we can obtain 49.5, and performance degrades over lesser ttt iterations as pointed out by the respected reviewer KZWi. We tried to discuss the computational efficiency of apm, and some baselines in table 4.
>
> ```On Questions: on components in fig 1 and their computations```
> Components in fig 1 refer to a new architecture called APM inspired from GLOM [10]. It contains a trigger column T to which a given image is routed. The computation it does is a folding-unfolding process to yield location-aware columns. Each column queries an MLP and decodes a location-specific feature. These features are gathered on the output via statistical-running-average to yield a representation. The final representation is used for zero-shot classification via textual-encoder of a teacher in the contrastive-space. More technicalities are discussed in our reply to the respected reviewer KZWi.
>
> Thank you so much for your time and we will be happy to respond to further clarifications.
>
> Sincerely,
>
> Authors

---

> ### Author Response · Authors · 2024-08-06
> **[Authors Reply] List of references used in our rebuttal to the respected Reviewer PHih**
>
> **List of References in Rebuttal**
>
> [R1] Diffusion-TTA: Test-time Adaptation of Discriminative Models via Generative Feedback
>
> [R2] Continual Test-Time Domain Adaptation
>
> [R3] Robust Test-Time Adaptation in Dynamic Scenarios
>
> [R4] Shu, Manli, et al. "Test-time prompt tuning for zero-shot generalization in vision-language models”.
>
> [R5] NOTE: Robust Continual Test-time Adaptation Against Temporal Correlation
>
> [R6] Robust Test-Time Adaptation in Dynamic Scenarios
>
> [R7] SoTTA: Robust Test-Time Adaptation on Noisy Data Streams
>
> [R8] Entropy is not Enough for Test-Time Adaptation: From the Perspective of Disentangled Factors
>
> [R9] Tent: Fully Test-time Adaptation by Entropy Minimization, ICLR 2021

---

> ### Comment · Reviewer_PHih · 2024-08-08
>
> While I have no doubt about the performance and efficiency of the APM, I have a serious concern that misunderstanding the terminologies and previous work's experiment settings may affect the reliability of the work. The misunderstanding implies that you never read any of TTT or TTA papers in detail and completely misunderstand their settings. My arguments are as follows:
>
> Response to "TTA adapts to the target distribution (using all test samples), whereas TTT adapts to each sample independently.
>
> This is completely wrong. Most TTA do not adapt to the whole test set. Many of them adapt to the batch samples or even to instance.
> Moreover, "TTT adapts to each sample independently" is completely wrong. It is unnecessary that TTT has to either adapt to one sample or the entire batch or test set. TTT/TTA lies on a special case of Domain Adaptation whereas the model has no access to the source data (source-free) but has to adapt to the target data during the test time.
>
> I suggest you read through all the papers about TTT and TTA. You will see some people in the community use TTA and TTT interchangeably [2]. I can see no significant difference between to the general setting between TTT and TTA.
> However, the pioneer like TENT [3] discusses a small difference between TTT and TTA. TTT requires the adjustment of training loss (adding an auxiliary loss term) while TTA only introduces the new loss during the test time.
>
>
> I can refer to "Network details" on section 3 of the pioneer test-time training (TTT) [1] (which some people also classify it as test-time adaptation [2]). They use different ResNets trained on ImageNet and CIFAR10. ImageNet-trained ResNet is evaluated on ImageNet-C and the same for CIFAR10. Regarding your method, as you use pretrained Vision Encoder as teacher, the vision encoder is already trained on ImageNet. This means your method also uses the knowledge from the training distribution extracted by the teacher model.
>
>
>
> [1] Test-Time Training with Self-Supervision for Generalization under Distribution Shifts
>
> [2] A Comprehensive Survey on Test-Time Adaptation under Distribution Shifts
>
> [3] Tent: Fully Test-time Adaptation by Entropy Minimization

---

> ### Author Response · Authors · 2024-08-08
> **Author's reply to the respected Reviewer PHih.**
>
> We thank the reviewer for their timely response to our rebuttal. We are grateful for the reviewer's insight and help in improving our work.
>
> We agree with the reviewer that the differences between TTA/TTT are indeed subtle/minor. As this difference do not play an important role in our contributions, we will refrain from differentiating between TTT/TTA works in our paper, and add more TTA works in related work and comparisons. We have shared comparisons with the recent TTA work aka Diffusion-TTA, and we will also add other such works.
>
> In our work, we have followed the experimental setup of TTT-MAE [NeurIPS 2022] and TPT [NeurIPS 2022]. Furthermore, we will be grateful to mention the reference to TENT [Neurips 2020] in TTT-MAE paper: "Other papers following [42] have worked on related but different problem settings, assuming access to an entire dataset (e.g. TTT++ [28 ]) or batch (e.g. TENT [48 ]) of test inputs from the same distribution. **Our paper does not make such assumptions, and evaluates on each single test sample independently**".
>
> Finally, we also point towards mention of TENT in the TPT paper: "However, TENT **needs more than one test sample** to get a non-trivial solution."
>
> We would like to clarify a few other points:
>
> - APM does not rely on any pretraining since weights are drawn from a normal distribution.
> While APM does rely on CLIP-pretrained weights, **we do not perform any training/pretraining on ImageNet**. We would be grateful to mention that CLIP was trained with 400M image-text pairs from OpenAI, and Openclip variant leveraged dfn5b. Furthermore, APM performs competitively on CLIP baselines and other methods which use CLIP weights.
>
> >> "Moreover, "TTT adapts to each sample independently" is completely wrong. It is unnecessary that TTT has to either adapt to one sample or the entire batch or test set"
>
> We agree with the reviewer here that TTT does not have to adapt to a single sample. Existing works like TTT-MAE[1] and others have focused on adapting to a single sample, as noted in their paper "By test-time training on the test inputs independently, we do not assume that they come from the same distribution".
>
> We will be happy to discuss any further concerns,
>
> Yours sincerely,
>
> Authors.
>
> [1] Gandelsman, Yossi, et al. "Test-time training with masked autoencoders." NeurIPS2022.
>
> [2] Shu, Manli, et al. "Test-time prompt tuning for zero-shot generalization in vision-language models." NeurIPS2022
>
> [3] Sun, Yu, et al. "Test-time training with self-supervision for generalization under distribution shifts." International conference on machine learning

---

> > ### Comment · Reviewer_PHih · 2024-08-08
> >
> > Thank you for the clarification. My concern has been addressed and I raised the score. This can be a good and interesting work after polishing writing for a few iterations.
> >
> > I have one more question, in the paper, you mention "Therefore, the MLP can be queried autoregressively."
> > What is the mathematical formulation of the autoregression you mentioned? Does the MLP produce the output based on the previous T?

---

> ### Author Response · Authors · 2024-08-08
> **Author's reply to the respected Reviewer PHih.**
>
> We thank the respected reviewer for their valuable time, and efforts in helping us to improve our work. We are grateful for the chance to engage deeply in the inner workings of APM.
>
> In our reply to the respected reviewer Kzwl ```On memory/states and auto-regression in APM```, we mentioned substituting the word autoregression with sequential, because the trigger column $T$ unfolds to yield location aware columns, i.e. $T_{ij} = (T|p_{ij})$, where $p_{ij}$ is the generated hard-coded positional encoding as being used in transformers[3], and neural fields[2]. These $T_{ij}$ columns are responsible for sequential querying of the MLP.
>
> Mathematically, the MLP predicts a location specific feature $f_{ij}=MLP(T_{ij})$ as a consequence of a forward-pass as each column gets pumped through the net. The MLP is queried ```sequentially``` with different columns $T_{ij}$ until the locations $ 1 \leq i \leq H$, $1 \leq j \leq W$ get exhausted, where $H,W$ are the dimensions of the input image $I$.
>
> In the shared code in supplemental, MLP does not contain explicit recurrence, because the trigger column $T_{ij}$ carries the **entire** sequence $T$. The $p_{ij}$ in $T_{ij}$ guides the MLP to decode location-specific feature $f_{ij}$, which is a form of feature-expression as also previously hypothesized in GLOM [1] (our changes over Section 2.1 and Figure 3 in the GLOM paper).
>
> We can also gain more depth by looking at Fig 3 of the GLOM [1]. The subtle differences are, 1) [a b] in that figure is the trigger column $T$ in APM, which carries the whole image $I$. 2)$x4$ is substituted by a positional encoding $p_{ij}$. 3) In addition to decoding location-specific RGB, we also decode higher dimensional features $f_{ij}$, for $ 1 \leq i \leq H$, $1 \leq j \leq W$ which allows APM to do field-based-decoding, which we leveraged for downstream-classification.
> We have also added a detailed pseudo-code for operation of APM during test-time-training in
> [Pseudocode](https://anonymous.4open.science/r/apm_rebuttal-F3D1/ttt_pseudocode.png).
>
> The behaviour of APM is then akin to neural fields[2] (as suggested by the respected reviewer doB6), for eg, neural fields fire iid rays into the MLP shared across all input rays and decode location-specific rgb. Similarly, APM shares a MLP across all columns and decodes location-specific features. While neural fields work for 3D view synthesis, APM works for 2D image perception, with potential for extensions to other higher-dimensional input-percept, as also duly noted in Sec 11 of the GLOM paper[1].
>
> As promised, we will also refine the paper-writing for better engagement with a prospective reader.
>
> Yours sincerely,
>
> Authors.
>
> [1] Hinton, Geoffrey. "How to represent part-whole hierarchies in a neural network." Neural Computation 35.3 (2023): 413-452.
>
> [2] Mildenhall, Ben, et al. "Nerf: Representing scenes as neural radiance fields for view synthesis." Communications of the ACM 65.1 (2021): 99-106.
>
> [3] Vaswani A, Shazeer N, Parmar N, Uszkoreit J, Jones L, Gomez AN, Kaiser Ł, Polosukhin I. Attention is all you need. Advances in neural information processing systems. 2017;30.

---

### Official Review · Reviewer_KZWi · 2024-07-13

**Soundness:** 2
**Presentation:** 3
**Contribution:** 2
**Rating:** 7
**Confidence:** 4

**Summary:**

This paper introduces a new algorithm for test-time training where the “test-time” task is overfitting for per-image CLIP / DINO feature distillation. The associated “downstream” task involves directly using the per-image network feature to perform image classification using dot product in CLIP space. Given an image, the method first proposes encoding it into a global feature vector via convolution. The global feature is then unfolded into per-pixel feature vectors by concatenating positional codes to the global features. These local features are then decoded and aggregated before being supervised with the CLIP image features distillation loss. This process is repeated for multiple iterations, as the global feature is “refined”. The final global feature turns out to be competitive or better than the base teacher model for some tasks and outperforms baselines. Despite the global CLS-token only supervision, the local bottleneck learns useful dense semantic features which are qualitatively visualized. The paper also shows some interesting qualitative findings when the method is trained across a dataset instead of on a per-image basis.

**Strengths:**

-	The paper shows that with their method, it is possible to recover dense pixel level features that are semantically meaningful (eg. features that recover object-ness, boundaries, etc) using only a global CLS token as supervision. I believe this finding is very interesting and has consequences for works even beyond the applications shown in the current paper (eg. for 2D -> 3D feature distillation)  From a representation learning perspective, this finding also reinforces the insight that the global CLS token contains useful geometric information about the image it encodes and that this information can be recovered using the right inductive biases in the distillation process.
-	The paper makes interesting biological analogies to GLOM to motivate it’s iterative “folding” and “unfolding” approach, and further connects it with its reasoning on using positional codes to break symmetry in the encoding process. Figure 3 shows that over time, the local features form islands of self similarity, as highlighted / predicted in GLOM’s model learning system. The result of learning local features and islands only from global supervision does indeed seem relevant to learning part-whole hierarchies in self-supervised tasks.
-	language-based zero-shot classification is the predominant metric chosen to evaluate the method and the paper does a thorough and sound job of running a wide range of experiments. The paper also compares with all representative sets of experiments, including a) directly using the teacher model b) other test time training approaches that optimize with a self-supervised task before prediction and c) prompt tuning approaches that are few-shot trained on the dataset. There is also thorough evaluation on a wide range of distribution shifted and cross distribution datasets. The method performs competitively and strongly on many of the benchmarks, and when it is outperformed by baselines, these runs are also duly noted, discussed and acknowledged.

**Weaknesses:**

I believe that the paper would benefit from some re-organization, clarification and writing improvements. At different points in the paper, there seems to be contradictory information regarding the nature of the method and some key details necessary to the success of the method are left out. Bulleted below:

-	In the method figure (Figure 1), a flattened rgb patch seems to be appended to the flattened image encoded with convolution, annotated $I_{xy}$. However, no references to this variable are made in the methods section. Further the methods section only mentions concatenating the global feature with the positional codes, without appending any patch specific information while querying the field.
-	In the same figure (Figure 1), the “folded state” in the yellow box seems to be the one that is fed into the MLP for decoding. However, in the methods section, it is claimed that it is the unfolded state that is sequentially fed into the MLP for decoding.
-	In the figure and on lines 65, 129, the method makes a reference to auto-regressive querying. I’m not sure what this means, as any way in which the model keeps “state” or “memory” is not clearly described. Further, it is not described how the output of any one MLP call is fed back into future steps. After looking at the appendix pseudocode, there does not seem to be any auto-regressive component. Was the intention to mention just sequential (and not auto-regressive) encoding?
-	Details about the unfolding process are clearly described, however details about the folding process are scarce. In Figure 1, the folding process seems to include a gather operation, while elsewhere it is described as an averaging operation. Further, the appendix pseudocode is lacking the “multi-iteration” nature of the algorithm.
-	When additional patch level feature losses are added (line 262) in the later sections, the details for how these are supervised are not clearly mentioned in writing.
-	The related work section needs to be significantly improved: it is presently under-cited and leaves out key related work in test-time training and prompting. These are mentioned and the differences described in other parts of the paper – but reorganization is required to improve readability. A further line of related work is in feature refinement / upscaling / distillation, and such papers need to be cited and compared to in order to elucidate differences and potential novelty in distillation method.

The paper contains several unsubstantiated claims that weaken the otherwise interesting results. Perhaps, they can be clarified better, or removed:

-	The paper claims that APM confirms that “percept is a field” as speculated by GLOM. While the paper does show that field-based decoding improves image classification, and that local features are emergent from global features, I believe a more general purpose evaluation of a broader range of perception tasks, including dense ones, must be done before this insight is claimed. As such, I believe this statement needs clarification as to what subset of ideas of GLOM are actually being “shown to work” in this paper.
-	The paper claims that APM doesn’t need dataset specific pre-training (Eg. line 176). While this is true in the sense that APM is a test-time instance method, APM does require access to highly meaningful visual representations (Eg. DINO / CLIP) that have been trained at scale. APM performance, as shown by the tables, degrades as the quality of the target representation degrades. I believe paraphrasing this claim is important to reflect this.

**Questions:**

I have several questions. Mainly important to clarification of the method:

-	APM compares to baselines such as Prompt ensembling in Table 1, which is a text query side improvement. Did authors similarly use the best performing “text prompts” for their method?

-	The per-instance distillation loss proposed by APM is minimized when the predicted feature is exactly the same as teacher feature. The fact that the final feature surpasses teacher feature performance implies that the loss does not go to 0 at training. Intuitively, what is the key constraint in the network that prevents loss from going to 0? What do the training curves look like for some example runs?

-	Summarizing the questions asked in weakness section 1:

o	Could you clarify what exactly does the folding process look like?
o	Is $I_xy$ (patch level info) appended to $T$ and fed into the decoding MLP?
o	Is the model actually auto-regressive or purely sequential / iterative?
o	What is the correct pseudo-code corrected with the 20 iterations? How is T (the encoded global feature) updated after one iteration of encoding?
o	How does performance on classification degrade with fewer iterations?

Answers to these questions are important for me to fully understand the paper.

**Limitations:**

The limitations section as currently written is a future work section. As mentioned in the “questions” section above, dense tasks and utility of local features is indeed interesting as a next direction. Additional things that can be discussed in the limitations section is the need for the method to do multiple iterations of test-time training instead of simple feed-forward inference or a single gradient step like other test-time training methods, and the variability in performance as compared to baselines.

---

> ### Author Rebuttal · Authors · 2024-08-06
>
> We thank the reviewer for giving us their valuable time. Please find our responses:
>
> ``` On using the local patch variable $I_{xy} in the fig1 , but not in the methods. ``` Fig 1 meant to be a general case of APM's operation. In methods, APM doesn't use $I_{xy}$. We were showcasing that positional codes break symmetry even without additional patch-prior and disentangle information from global $I$. Ablations in Tab 5b), validate local-patch-injection improving from 96.5-> 96.8 on C-10 .
>
> ``` On Whether unfolded state is being feed-forwarded through mlp or the folded state?``` Trigger column T exists in the folded state after a fwd pass(fig 1b), when knowledge has 'collapsed' into the weights of conv layer. When fwd-pass begins, T opens up (unfolds), and 'all' columns are forwarded through  mlp. We apologize, since the middle column in fig 1a) is shown as folded and lent itself to confusion. We will update it.
>
> ```On memory/states and auto-regression in APM``` APM encodes memory in 1) weights of T, and 2) weights of MLP. **Both** T and MLP are shared across **all** the locations. Weight-sharing induces the distributed-memory to learn relevant synaptic-patterns [11].
>
> Auto-regression unrolls a shared-decoder over time. In contrast, APM holds the whole sequence in $T$, and directly hops onto a space/time-step [R1] via a location-column. Recurrence/feedback-loops are compensated for by a form of feature-expression ([12] & our changes over GLOM [10] sec 2.1, para 2. Positional code gives the notion of distance from starting point T, and iid columns help ensure parallel-processing of columns as in the third line of fig1 caption).
>
> We convey this by choosing ```sequential```. We will replace `auto-regression` in fig1a) and line 65.
>
> ```On Details of folding-process/the gather step``` Folding operates at ```input``` of APM at the ```end``` of fwd-pass. Since a column $T_{ij}= (T|p_ij)$, and $p_{ij}$ is governed by hard-coded sinusoids, each $p_{ij}$ can be thought of as being ```annihilated``` during folding[14-16], aka deleting $p_{ij}$. Gradients from all $T_{ij}$ then flow back in towards the (center-of-mass[R3]/reservoir[R4]) column T[10].
>
> 'Gather' happens during the unfolded phase ```at mlp output```:  for a particular generated column $T_{ij}$, the net is predicting the feature $f_{ij}$. Gather estimates statistically running average of $f_{ij}$, as different $T_{ij}$ get  pumped. Gather's output is used for final classification.
>
> ``` On Sources of patch-level features during supervision```: We will clarify that patch-level features are the last-layer of a teacher and supervised via $L_{2}$ constraint even further in ablation section(line 231).
>
> ```On APM's claims```:**percept is a field**: We will rephrase as "APM is a step towards validating if input percept is a field". We will revise line 176 as: apm can work with randomly initialized weights, but requires a single representation distilled from a model trained at scale.
>
> ```On apm's reliance on a teacher```: We performed an additional experiment by removing the teacher from the apm, and doing rgb reconstruction on coco-train. L_2 rgb loss on coco-val fell to 0.0027. Training took far longer than if feature-vectors were also distilled from a teacher.
> Higher-dimensional vector-spaces carry more bits[R5], but incur significant randomness. Supervision from a stronger teacher compensates and guides apm to correct points in subspace. Progressing from vit b->h in tab1, makes apm more competitive.
>
> We will add limitation: a.k.a reduced performance as cls token's dimension reduces.
>
> ```On Using prompt-ensemble on the textual side```: apm's imagenet-experiments in tab 1 used 80 hand-crafted prompts, same as CLIP. Openclip vit-h baseline also used the prompt ensemble, where apm also obtains competitive performance.
>
> ```On Intuition behind final apm features surpassing teacher, stopping constraints```: APM relies on distillation, where students have been observed to outperform their teachers. In fig 3,  over 250 ttt iterations, the loss dropped from 1e-3 to 1e-12. We ```had to``` reduce the learning rate by a factor of 0.1 every 50 iterations. Whilst small, indeed it is not perfectly zero. One reason might be finite 64-bit precision of neural synapses and low learning rates. We ```waited-long``` to creep gradients into the net, as in  [11]'s footnote 20.
>
> One might build additional constraint in a student (aka APM) to estimate when its own fantasies[18]/predicted semantic-features are better than the teacher's and stop dynamically [R6]/recurse. Notions on `soft` decision-making for higher-level cognition are embedded in [R4,R7].
>
> ```On Multi-iteration pseudo-code for ttt```: Please find an anonymous code-link[Pseudocode](https://anonymous.4open.science/r/apm_rebuttal-F3D1/ttt_pseudocode.png). We will update the paper.
>
> ```On Quantitative effects of lower number of ttt iterations```
> We provide the ablation in global rebuttal pdf in Table 1.
>
> ```On Improving related work```
> We will discuss more ttt/prompting/tta approaches. We shall add differences w.r.t feature refinement/distillation: a.k.a $L_2$ mimicking feature grids vs boltzmann-temperature-matching logits[3].
>
> ```On Extending limitations section```We apologize if our limitations came across as optimistic future work. We have noted APM's limitation: APM requires multiple iterations for test time training, which might increase inference time in time-sensitive scenarios. It would be very interesting to evaluate its performance leveraging other zero-th order optimization techniques [11].We also report mean/std of ttt on dtd with variable its/seeds in global pdf. We also provide more results on imagenet-c  with additional noise severities (1-4).
>
> ```On Adding future work``` We will add apm's potential on dense tasks.
>
> Yours Sincerely,
>
> Authors
>
> p.s please find references in comments

---

> ### Author Response · Authors · 2024-08-06
> **[Authors Reply] List of references used in our rebuttal to the respected Reviewer KZWi**
>
> p.s. other references follow paper order.
>
> Yours Sincerely,
>
> Authors
>
> R1: Einstein, Albert. "On the electrodynamics of moving bodies."
>
> R2: Geoff Hinton at Stanford, YouTube video, timestamp 49:07, https://www.youtube.com/watch?v=CYaju6aCMoQ
>
> R3: LA Homogeneous Universe of Constant Mass .
>
> R4: Neumann, John. "Theory of self-reproducing automata."
>
> R5: Shannon, Claude Elwood. "A mathematical theory of communication."
>
> R6: Hinton, Geoffrey E., et al. "The 'wake-sleep' algorithm for unsupervised neural networks."
>
> R7: Bengio, Yoshua. "The consciousness prior."

---

> > ### Comment · Reviewer_KZWi · 2024-08-11
> >
> > Thank you for your rebuttal. My main concern with the paper (reason for borderline score) was the lack of clarity regarding the technical details of the folding / unfolding . distillation processes  and the lack of clarification / phrasing of certain claims regarding inspirations / analogies in the paper. The authors have sufficiently addressed the technical questions, clarified intuition and promised to incorporate the writing changes in the final draft of the paper. These promised changes would be important for a strong final paper.
> >
> > With my questions clarified, I will increase my score.

---

> > > ### Author Response · Authors · 2024-08-11
> > > **Author's reply to the respected reviewer KZWi**
> > >
> > > We thank the respected reviewer KZWi for their kind consideration of our work. As promised, we will incorporate the writing comments and the reviewer feedback to improve the engagement of a prospective reader.
> > > Thank you so much for your valuable time, and we will be grateful to respond to any further clarifications,
> > >
> > > Yours sincerely,
> > >
> > > Authors.

---

### Official Review · Reviewer_doB6 · 2024-07-13

**Soundness:** 3
**Presentation:** 3
**Contribution:** 4
**Rating:** 7
**Confidence:** 3

**Summary:**

The authors propose a new test time training method (architecture + self-supervised task) called Asynchronous Perception Machines (APMs). APM is computationally efficient, and empirically matches or improves performance on OOD image classification tasks versus prior test time inference approaches. The approach considers each $c' \times h \times w$ feature vector in the output of an image encoder as a c' dimensional mapping of the input spatial features to this 'island of agreement'. A convolutional neural network is used to generate this from the image. Once generated, this location aware column is folded/unfolded at inference time based on the input patch, and used to reconstruct the RGB and location specific features. The features are then averaged and compared against a text feature vector to perform classification. APM processes image patches one at a time in any order, thus also exploiting the benefits of transformer style models. Empirical results show that APM does well on out-of-distribution image classification benchmarks and exhibits competitive performance compared to existing TTT.

**Strengths:**

- The method proposed is very novel, and (in my knowledge) the first practically viable and useful approach towards exploiting object part heirarchies as they were proposed in the GLOM paper. There are a lot of moving parts, which may or may not sustain in future work, but the key ideas (collapsed/expanded feature vectors, contrastive objective for tti instead of manually defined pre-text task) all seem like steps in the right direction.
- The authors should be commended for reproducible science, reading through the code and instructions accompanying the paper makes it easier to understand the novel model better. It also helps establish the credibility of their work, and since this is a nascent topic will engender future research.

**Weaknesses:**

- While I think test time training is an important and useful paradigm, the illustrative example of "self-driving car trying to make sense of the visual world while it is raining, but there were no such training scenarios in its training data" (L20-21) does not really apply because tts does not really allow for instantaneous decision on a new test instance.
- I don't think the analogies in 3.1 are anything beyond that, and as such don't belong in the main paper. It would be better to draw real scientific analogies in the main paper, e.g. to 3D novel view synthesis.

**Questions:**

Typo L17: Neural can now -> Neural nets can now

**Limitations:**

Discussed briefly in Section 8

---

> ### Author Rebuttal · Authors · 2024-08-06
>
> Thank you for your time and efforts in evaluating our work and providing positive and constructive feedback. We will be happy to address your comments:
>
> ```Illustrative example, “self-driving car trying to make sense of the visual world while it is raining, but there were no such training scenarios in its training data”```
>
> We thank the reviewer for this insight, and we will replace this with a better example. We agree that TTT decisions are not made ```instantaneously``` in one iteration, in the cases where the net was not-pretrained on any source data, which also happens to be the setting where APM was evaluated. Some of the experiments did showcase APM being able to semantically cluster on COCO-val, after training on large datasets like COCO-train (Fig 5). It would be an interesting future-work to see optimizations such source-training brings towards reducing required ttt-iterations. APM is also computationally efficient (tab 4) compared to other baselines. It would be very interesting to evaluate its performance leveraging other zero-th order optimization techniques [R1,R2]. As [R3] also points out, such efficient-optimization would indeed be ```a good application``` for helping make instantaneous-decisions. We thank the reviewer for this observation, and will discuss this in future work also.
>
> ```Analogies in sec 3.1```
> Thank you for this comment. As also pointed out by multiple-reviewers, we will shift these analogies to supplementary. We thank the reviewer for the suggestion to link our insights to 3D novel view synthesis. Indeed, given a single 3D-spatial coordinate of a pin-hole camera, neural fields operate by shooting i.i.d rays into the scene and leveraging the MLP to decode rgb. In a similar way, APM leverages location independent columns, a shared MLP, to decode location-specific features. While neural fields work for a 3D scene, apm works for a 2D percept, on a collection of images. We will add such a technical-analogy to our paper.  We will correct the spelling mistake on line 17, and add the word 'nets'.
>
> We thank the reviewer for giving us their valuable time, and will be grateful for the chance to engage in further discussions.
>
> Sincerely,
>
> Authors
>
>
> [R1] Hinton, Geoffrey. "The forward-forward algorithm: Some preliminary investigations." arXiv preprint arXiv:2212.13345 (2022).
>
> [R2] Malladi, Sadhika, et al. "Fine-tuning language models with just forward passes." Advances in Neural Information Processing Systems 36 (2023): 53038-53075.
>
> [R3] Sun, Yu, et al. "Learning to (learn at test time): Rnns with expressive hidden states." arXiv preprint arXiv:2407.04620 (2024).

---

> > ### Comment · Reviewer_doB6 · 2024-08-14
> > **Response to Rebuttal**
> >
> > I'd like to thank the authors for taking the time to respond to the review, as well as for accepting some of my suggestions regarding the paper. After reading through the general rebuttal, as well as the author's rebuttals to each of the reviews - I agree with other reviewers that the writing can be improved (especially in terms of clarity and organization). This can be done in the camera ready, and as such I think the APM paper is good enough to appear at NeurIPS. I would like to keep my original rating of Accept, and look forward to seeing this work at NeurIPS.

---

> > > ### Author Response · Authors · 2024-08-14
> > > **Author's response to the Respected Reviewer doB6**
> > >
> > > We thank the respected reviewer doB6 for their kind words with regards to APM. We will further improve the paper's clarity and organization for better engagement with a prospective reader.
> > >
> > > We remain,
> > >
> > > Yours sincerely,
> > >
> > > Authors.

---

### Official Review · Reviewer_X1eH · 2024-07-13

**Soundness:** 2
**Presentation:** 1
**Contribution:** 2
**Rating:** 5
**Confidence:** 3

**Summary:**

This paper proposes a computationally-efficient architecture for test-time-training. APM can process patches of an image in any order asymmetrically, where it learns using single representation and starts predicting semantically-aware features. The experiment results demonstrates the effectiveness and efficiency of the method.

**Strengths:**

The experiments on zero-shot test-time-training have demonstrated the effectiveness of the proposed method.

**Weaknesses:**

1. The writing can be improved. The current organization makes me feel very confused of the proposed method. For example, in the abstract, the novelty should be emphasized. It seems that processing patches of an image one at a time in any order asymmetrically is emphasized, which I don't think it is a novel contribution.

2. As this paper targets at solving problems in test-time-training, I think more related work in this domain should be introduced in the section of related work.

3. I cannot understand why the analogy compared with biology and computation makes sense. The authors introduced a lot of analogy, which has no proof and no explanation. I think the technical soundness of the paper should be significantly improved.

4. The training and test details in the experimental section should be demonstrated in details, as readers may be interested in the experimental setting.

5. I am wondering if the proposed method can be extended to few-shot testing scenarios?

**Questions:**

See weakness.

**Limitations:**

See weakness.

---

> ### Author Rebuttal · Authors · 2024-08-06
>
> We thank the reviewer for helping us improve our work. Please find our responses below:
>
> ```[1] On processing patches of an image one at a time/lack of novelty```
>
> We apologize that the novelties of our work were not clear. A CNN filter in any layer could process any patch by directly sliding on that region and performing convolution. Therefore, it can be said to still do asymmetric processing. However, for features to flow in the next layer, the whole CNN filter operation in the current layer will need to be finished thereby raising a waiting issue.
>
> APM improves upon this by 1) breaking symmetry by location-aware columns, and 2) layer skipping: which allows apm to learn a mapping from input pixels to the last layer feature of a model via co-distillation. We will highlight these aspects more in the abstract.
>
> Furthermore, as pointed out by reviewer KZWi, we will also focus on the ability of apm to recover scene-semantics from a global cls token and its potential for learning part-whole hierarchies via ssl-tasks.
>
> ```[2] On improving related work by adding more works``` We agree that our related work could be improved with works from other relevant areas such as test-time-adaptation as mentioned by reviewer PHih. Furthermore, as pointed out by reviewer KZWi, we will add more  prompting approaches. We have also made a note to add an additional section clarifying the feature refinement/distillation.
>
> ```[3] On the presented biological analogy in section 3.1``` We apologize if our biological analogy motivated by GLOM came across as non-technical. We will shift biological insights (sec 3.1) to supplementary. We have provided more clarifications on trigger columns, folding-unfolding operation in our responses to  reviewer KZWi. We have also added pseudo-code illustrating apm's operation during TTT (https://anonymous.4open.science/r/apm_rebuttal-F3D1/ttt_pseudocode.png).
>
> ```[4] On providing precise-hyperparameters/train/test-details for reproduciblity ``` Taking inspiration from [38], we provide **Reproducibility Statement**: In order to ensure the reproducibility of our experiments, we have shared the code in supplementary during the review process. The code, model weights shall be released publicly post-review. We will also share a docker image.
>
> Hyperparameter tables: Code has been shared in supplemental.
>
> | Hyperparameter           	| Value                                                                              	|
> |------------------------------|----------------------------------------------------------------------------------------|
> | **Number of Test Samples**   | 50000 (Imagenet Splits), variable for other datasets.                               	|
> | **Testing Iterations**   	| 20                                                                                 	|
> | **Batch Size**           	| 1                                                                                  	|
> | **Learning Rate**        	| 1e-4                                                                               	|
> | **Optimizer**            	| Adam                                                                               	|
> | **Feature Output Size $d$**  | $768/1024$                                                                         	|
> | **Positional Encoding Size** | $768/1024$                                                                         	|
> | **Image/Crop Size**      	| 448                                                                                	|
> | **Augmentations**        	| Normalization,  $\mu = (0.485, 0.456, 0.406)$, $\sigma= (0.229, 0.224, 0.225)$     	|
> | **Precision**            	| fp16 (grad-scaled)                                                                 	|
> | **Num of Workers**       	| 8                                                                                  	|
> | **Operating System**     	| 1x rtx a6000 48GB/96GB RAM/Ubuntu 22.04/2TB SSD/5TB HDD                            	|
>
> Architecture details:
> With input dimensions \(h, w, c\) and feature dimension \(d_p\): dimensionality of positional encoding. \(s\): stride of convolutional filter in encoder, \(d_c\): dimension of the CLS token of the teacher on which APM learns.
>
> | Component        | Layer  | Feature Dimension   | \(n_{\text{kernels}}\) | Stride | Padding     	|
> |--------------------------|--------|--------------------------|-----------------------|--------|-----------------|
> | **Input**            	|    	| \(h X w X c\)  | -                 	| -  	| -           	|
> | **Encoder**          	| Conv   | \(h_s X w_s X d) | 1   | \(s\)  | 0 / 0       	|
> | **Decoder**          	| Linear | \((d_p + d) X 4096\) | -                 	| -  	| -           	|
> |                      	| Linear | \(4096 X 4096\) 	| -                 	| -  	| -           	|
> |                      	| Linear | \(4096 X 4096\) 	| -                 	| -  	| -           	|
> |                      	| Linear | \(4096 X 2048\) 	| -                 	| -  	| -           	|
> |                      	| Linear | \(2048 X 1024\) 	| -                 	| -  	| -           	|
> | **Feature Projection Head** | Linear | \(1024 X d_c\) 	| -                 	| -  	| -           	|
> | **RGB-Head (optional)**  | Linear | \((d_p + d + 1024) X 4096\) | -        	| -  	| -           	|
> |                      	| Linear | \(1024 X 3 X 256\) | -             	| -  	| -           	|
> |                      	| Linear | \(256 X256\)   	| -                 	| -  	| -           	|
> |                      	| Linear | \(256 X 3\)     	| -                 	| -  	| -           	|
>
> ```[5] On extensions of apm to few-shot testing cases```
> We appreciate the reviewer's forward-outlook of apm's application beyond ttt (line 216 of paper). We will add an additional discussion on few-shot-testing in future work.
>
> We will be happy to respond to further clarifications. Thank you so much for your valuable time,
>
> Yours Sincerely,
>
> Authors
>
> p.s. references follow paper order

---

> > ### Comment · Reviewer_X1eH · 2024-08-10
> >
> > Thanks for your rebuttal. Most of my concerns are well addressed, so I increased my score.

---

> > > ### Author Response · Authors · 2024-08-10
> > > **Author's response to the Respected Reviewer X1eH**
> > >
> > > We thank the respected reviewer for their timely response to our rebuttal. We are grateful for the reviewer's consideration and will be happy to respond to further clarifications.
> > >
> > > Yours sincerely,
> > >
> > > Authors.

---

### Author Rebuttal · Authors · 2024-08-06

Dear Reviewers,

We appreciate the positive feedback from the reviewers for our work. The reviewers acknowledged several aspects such as the creativity [Reviewer PHih], novelty [Reviewer doB6], effectiveness [Reviewer X1eH], first practically-viable approach towards GLOM's [10] ideas [Reviewer doB6], being reproducible [Reviewer doB6], being able to recover semantics from just one global CLS token [Reviewer KZWi], potential for applications beyond TTT for general-representational-learning [Reviewer KZWi], exhaustive evaluations/comparisons across multiple benchmarks [Reviewer KZWi] and demonstrating competitive-empirical results [Reviewer KZWi].

Based on reviewers constructive-feedback, we make further positive-amendments to our work;

 -  We provide a global rebuttal pdf, containing ablations with variable ttt-iterations[Reviewer KZWi], 4 additional tables with varying noise-severity levels on imagenet-c across all $15 \times 4=60$ noises, with corrections in table-columns duly noted in captions[Reviewer PHih]. We will update the main paper tables ( Tab 1-3).
-  In addition to the code shared in supplemental, we add reproducibility statement, additional table for hyperparameters,  a new table describing apm's architecture in comments. We shall also release a docker-image [Reviewer X1eH] for computing machines with differing hardware-configurations.
- We have anonymously added ttt pseudo-code as an external link [Pseudocode](https://anonymous.4open.science/r/apm_rebuttal-F3D1/ttt_pseudocode.png)[Reviewer KZWi].

We shall update the supplemental accordingly

**Addressing paper-writing comments**

-  We will shift APM's biological-analogies (sec3.1) in supplemental [Reviewers X1eH,doB6], and clarify potential-novelties even better in abstract [Reviewer X1eH].
- We will extend related-work/limitations/future-work sections to tenth-page of the manuscript, and discuss more (ttt/prompting-approaches/tta/knowledge-distillation approaches [All reviewers] in related work), (applications to few-shot-testing [Reviewer X1eH], dense-tasks [Reviewer KZWi], using apm's local-field [Reviewer KZWi] in future-work).

We thank the reviewers for giving our work their valuable time. We have also provided individual-responses and look forward to the chance to engage in further-discussions.

Yours Sincerely,

Authors.

---

### Decision · Program_Chairs · 2024-09-25

**Decision:**

Accept (poster)

**Comment:**

The paper proposes a novel test-time training method, Asynchronous Perception Machines (APMs), which is computationally efficient and demonstrates robust performance on out-of-distribution image classification tasks. The method leverages patch-based processing and knowledge distillation to improve classification accuracy while requiring fewer computational resources. Strengths include its novel approach to handling test-time training and the clear empirical validation of its effectiveness. The method’s ability to process image patches in any order and its competitive performance across benchmarks are notable. Please address reviewers’ comments in the final version.